# A Long $N$-step Surrogate Stage Reward for Deep Reinforcement Learning

**Junmin Zhong**
Arizona State University

**Ruofan Wu**
Arizona State University

**Jennie Si** *
Arizona State University

## Abstract

We introduce a new stage reward estimator named the long $N$-step surrogate stage (LNSS) reward for deep reinforcement learning (RL). It aims at mitigating the high variance problem, which has shown impeding successful convergence of learning, hurting task performance, and hindering applications of deep RL in continuous control problems. In this paper we show that LNSS, which utilizes a long reward trajectory of future steps, provides consistent performance improvement measured by average reward, convergence speed, learning success rate, and variance reduction in $Q$ values and rewards. Our evaluations are based on a variety of environments in DeepMind Control Suite and OpenAI Gym by using LNSS piggybacked on baseline deep RL algorithms such as DDPG, D4PG, and TD3. We show that LNSS reward has enabled good results that have been challenging to obtain by deep RL previously. Our analysis also shows that LNSS exponentially reduces the upper bound on the variances of $Q$ values from respective single-step methods.

## 1 Introduction

Reinforcement learning (RL) shows great promise as a theory of learning in complex dynamic tasks [1, 2]. For continuous control problems involving high-dimensional states and control spaces, several deep reinforcement learning (DRL) algorithms, such as deep deterministic policy gradient (DDPG) [3], proximal policy optimization (PPO) [4], Soft actor critic (SAC) [5], D4PG [6] and twin delayed DDPG (TD3) [7], have demonstrated their great potential.

However, the high variance problem [8, 9] in DRL still poses great challenges for it to be effective in solving complex contentious control problems. On one hand, this problem is intrinsic to the trial and error nature of reinforcement learning, which causes low data efficiency, slow learning, or even instability of learning and poor task performance [10, 11]. On the other hand, high variances are often caused by several factors, such as variances in value estimation [12], variances resulting from different random seeds [10], and variances due to noisy stochastic reward or corrupted sparse reward [13]).

Different approaches have been proposed to address the high variance problem, from the original TD($\lambda$) algorithm to $Q(\sigma)$ [14] as a generalization of TD($\lambda$) [15]. Rollout methods [16, 17, 18] have enabled the success of some of the most important milestone cases such as Backgammon [17] and AlphaGo [19] in machine learning history. Although these algorithms has been demonstrated in discrete state and control problems, variants of rollout are believed to be effective also in continuous control tasks. Modern DRL algorithms such as Rainbow [20] integrate $n$-step learning on top of DQN and results in a speed-up in learning and an improvement of the final performance on Atari 2600 games. D4PG [6] uses the same $n$-step methods to update target values, and have been shown effective in continuous control tasks. PPO [4] uses a truncated version of generalized advantage estimation (GAE) to update policy which reduces policy variance [21]. However these methods do

---

*Corresponding author: si@asu.edu

37th Conference on Neural Information Processing Systems (NeurIPS 2023).

not explicitly account for noisy rewards where such uncertainties have been reported to degrade the model/controller performance [22].

In this paper, we propose a new algorithm, namely the long $N$-step surrogate stage (LNSS) reward estimator. LNSS aims at providing variance reduction for DRL methods in continuous control problems, even under noisy reward signals or corrupted sparse reward. LNSS is simple, and can be easily piggybacked on DRL algorithms. It is based on a surrogate reward $r'$ obtained from a trajectory of rewards of future "$N$-step"s. In turn, $r'$ is used in $Q$ value update where the target function is then of the same form as that in single-step TD methods. Using a long "$N$-step"s of the reward sequence containing environment information, it is expected to be more advantageous to learning than using a short trajectory of $n$-step of the reward sequence.

We show theoretically that the LNSS method exponentially reduces the upper bound on the variance of $Q$ value from a respective single-step method. We then conduct extensive experiments in selected environments in DMC and GYM where most current DRL algorithms such as DDPG, D4PG, and TD3 that typically struggle to obtain good results [23]. We consider uniform random noises of different magnitudes to show how LNSS can benefit a variety of DRL algorithms while the performances of other methods degrade significantly. We further show how LNSS can still be effective in environments where only sparse rewards are available.

**Contributions**. 1) We introduce a new, simple yet effective surrogate reward $r'$ that interpolates between the single-step TD methods and Monte-Carlo (MC) methods [15], to trade off the strengths from the two extremes. 2) We theoretically show the proposed method helps reduce the upper bound on the variance of $Q$ value with a long trajectory of rewards of future "$N$-step"s. 3) Extensive experiments on DMC and GYM benchmarks show that LNSS robustly benefits the learning performance of a variety of DRL algorithms even under reward noise or sparse reward. Mainly, the LNSS consistently reduces variances in the learned value function and achieves robust results from different random seeds in several different types of continuous control problems. Please check our code

## 2 Related Work

**The "$n$-step" methods**. Here, as in the literature, $n$-step methods refer to those involving a trajectory of rewards of multiple future steps in DRL algorithms for speeding up learning and reducing variance [24, 25, 6]. They usually proceed as follows. First, collect $n$-step returns to form a discounted sum of rewards [26, 27]. Then, perform $n$-step return backup to update the value function [15]. It was empirically suggested that $n$-step methods perform better than single-step methods since $n$-step returns propagate rewards to relevant state-action pairs more efficiently [24, 25].

Using longer trajectories with a larger $n$ is expected to improve learning performance [28]. However, to date, the $n$-step DRL methods have only been demonstrated by using relatively small $n$. Rainbow [20] uses $n = 1, 3, 5$ to evaluate performance of 57 Atari 2600 games and reported $n = 3$ as the best-performing reward trajectory length. D4PG [6] uses $n = 1, 5$ to evaluate a variety of continuous control tasks in DMC and results show that $n = 5$ performs uniformly better than others. However, 3 and 5 are relative small for complex tasks which usually have a maximum of 1000 steps in an episode.

There are potentially two reasons that prevent $n$-step methods from using larger length $n$. One is that the longer the $n$-step return, the larger reward scale that can cause saturation and inefficiency in learning for most DRL algorithms [10]. The second reason is that the longer the $n$-step return, the smaller discount factor ($\gamma^n$) in $n$-step backup [15]. Based on previous studies [29], variations in the scale of the discount factor directly causes uncertainty to the RL performance. New algorithmic ideas are needed for DRL algorithms to benefit from longer reward bootstrapping.

**Reward estimation**. Reward estimation aims at reducing sample complexity and learning variance as reward variances may be introduced from using random seeds in training [9], from sensor noises in real-world applications [8] and from corrupted sparse rewards [13]. Several approaches have been proposed for reward estimation. Small backups [30], instead of directly using an actual reward, use multi-step imaginary rollouts as a predicted reward. Model-based policy search with Bayesian neural network [31] finds expected cost by averaging over multiple virtual rollout. The $\lambda$-prediction [32] uses an extra network to predict reward and the training of which may introduce additional variance. Model-based value expansion [33] uses an imaginary reward from a model to estimate the values of state value functions. To reduce learning variance resulted from noisy rewards, regression reward

estimator [13] expects the estimated reward to reduce the variance propagated to the value function. Unbiased reward estimator [22] estimates a surrogate reward by learning a reward confusion matrix which estimates true reward from noisy reward. However, all these estimation methods require extra models (either in the form of neural networks, environment transitions, or reward noise model), which imposes overhead on learning and introduces estimation error. Additionally, how to effectively and efficiently integrate these reward estimates into DRL algorithms is not clear, or at least not straightforward.

**LNSS**. In contrast, our method does not introduce any new hyper parameters into learning as we still use the same form of reward signals but only to a longer horizon of "$N$-step"s. The LNSS method can be piggybacked on any DRL algorithms directly. Comparing to $n$-step methods, by using our surrogate reward $r'$ to perform single-step updates, we can benefit from $n$-step returns without changing the reward scale. This frees LNSS from a limited reward horizon to a much longer length $N$.

**Selecting DRL algorithms**. First, note that few existing $n$-step methods have been successfully applied to different DRL algorithms with consistent learning variance reduction. We selected DDPG, D4PG, and TD3 as base DRL algorithms, with which our LNSS is to be piggybacked on for the following considerations. Among all DMC based benchmark results [23], distributional method D4PG [6] outperforms other DRL algorithms such as DDPG [3], and TD3 [5]. On the other hand, TD3, DDPG have relatively good performance in simple tasks such as walker but struggled to achieve good results in harder DMC tasks such as humanoid-walk and fish-swim. We thus use the three chosen DRL algorithms to systematically test the idea of LNSS piggybacked on promising DRL algorithms in complex tasks, even in noisy reward environments.

## 3 Background

**Reinforcement Learning**. We consider a reinforcement learning agent interacts with its environment in discrete time. At each time step $k$, the agent observes a state $s_k \in \mathcal{S}$ and select an action $a_k \in \mathcal{A}$ based on its policy $\pi : \mathcal{S} \to \mathcal{A}$, namely, $a_k = \pi(s_k)$, and receives a scalar reward $r(s_k, a_k) \in \mathcal{R}$ (use $r_k$ as short hand notation).

Evaluation of a policy $\pi$ is performed using the expected return after taking an action $a_k$ based on state $s_k$ following the policy $\pi$:

$$Q^\pi(s_k, a_k) = \mathbb{E}[R_k | s_k, a_k]$$
$$\text{where } R_k = \sum_{t=k}^{\infty} \gamma^{t-k} r_t,$$
$$s_k \sim p(\cdot \mid s_{k-1}, a_{k-1}),$$
$$a_k = \pi(s_k),$$

(1)

with $0 < \gamma < 1$. The two common approaches (single-step and $n$-step) to update the $Q$ value for a policy $\pi$ are as follows.

**Single-step RL Algorithms**. RL algorithms rely on using TD error or residual of the Bellman equation to update the state-action $Q$ value for a policy $\pi$, as described below,

$$Q^\pi(s_k, a_k) = r_k + \gamma Q^\pi(s_{k+1}, a_{k+1}). \tag{2}$$

This equation provides the basis for single-step RL methods. It is also known as a backup operation since it transfers information from one step ahead back to the current step.

**"$n$-step" RL algorithms**. Using $n$-step rewards for faster reward propagation in RL has long been investigated [26, 14, 6, 20]. In $n$-step methods, the value function $Q^\pi(s_k, a_k)$ update using an $n$-step return is based on the following,

$$Q^\pi(s_k, a_k) = \sum_{t=k}^{k+n-1} \gamma^{t-k} r_t + \gamma^n Q^\pi(s_{k+n}, a_{k+n}). \tag{3}$$

The $n$-step return is expected to help agents learn more efficiently by affecting multiple state action pairs within one update and gain information over the $n$-step horizon.

More details on specific modern DRL algorithms such as the single-step methods (DDPG and TD3) and the $n$-step methods (D4PG and PPO) can be found in Appendix D.

## 4   Long $N$-step Surrogate Stage (LNSS) Reward

In this section, we introduce LNSS based on infinite horizon discounted reward formulation of reinforcement learning. Given a reward trajectory of $N$ steps from time step $k$, let $G(s_{k:k+N-1}, a_{k:k+N-1}) \in \mathbf{R}$ (use $G_k$ as short hand notation) denote the discounted $N$-step reward, i.e.,

$$G_k = \sum_{t=k}^{k+N-1} \gamma^{t-k} r_t, \tag{4}$$

where $r_t$ is the $t$th stage reward and $t$ is from $k$ to $k+N-1$. In LNSS, we introduce $r'_k$ as a surrogate stage reward in place of $r_k$ in Equation (2). To determine $r'_k$, we treat it as a weighted average of the $N$-step reward sequence, namely

$$r'_k = \frac{\sum_{t=k}^{k+N-1} \gamma^{t-k} r_t}{\sum_{n=0}^{N-1} \gamma^n}. \tag{5}$$

We then propose the surrogate stage reward $r'_k$ to be

$$r'_k = G_k \frac{\gamma - 1}{\gamma^N - 1}. \tag{6}$$

This surrogate stage reward $r'_k$ as formulated in Equation (6) relies on a discounted reward of an $N$-step horizon, from time step $k$ to step $(k + N - 1)$, from the stored experiences into a temporary replay buffer $\mathbb{D}'$. For a training episode of $T$ steps $[0, 1, 2..., T]$, the $\mathbb{D}'$ is a moving window of size $N$ from the initial state $s_0$ until the terminal state $s_T$. As a result, when there is a sufficient number (i.e., $N$) of reward samples, LNSS computes the surrogate reward $r'_k$ from below,

$$r'_k = \frac{\gamma - 1}{\gamma^N - 1} \sum_{t=k}^{k+N-1} \gamma^{t-k} r_t. \tag{7}$$

Note that, when the reward is estimated at the beginning or toward the end of a long trial, less than $N$ reward samples are available for estimating $r'_k$. A simple adjustment is given in Appendix C.

Once $r'_k$ is obtained, $r'_k$ and state action pairs $(s_k, a_k, s_{k+1})$ will append as a new transition $(s_k, a_k, r'_k, s_{k+1})$ stored into the memory buffer $\mathbb{D}$.

Note that many DRL algorithms [4, 20, 6] use distributed learning procedure to accelerate experience sample collection. We use the same technique to speed up sampling experiences. Then a DRL algorithm is ready to update the $Q$ value and the respective policy based on mini-batch data from the memory buffer. In general form, we have

$$\begin{aligned} Q_{i+1}(s_k, a_k) &= r'_k + \gamma Q_i(s_{k+1}, \pi_i(s_{k+1})), \\ \pi_i(s_k) &= \arg\max_{a_k} Q_i(s_k, a_k), \end{aligned} \tag{8}$$

where $i$ is iteration number. Putting the above two equations together, we have

$$Q_{i+1}(s_k, a_k) = r'_k + \gamma \max_{a_{k+1}} Q_i(s_{k+1}, a_{k+1}). \tag{9}$$

**Remark 1**. 1) Different from $n$-step methods, in our LNSS, $N$ is the number of steps for accumulating rewards in Equation (6. We still perform a single-step update as in Equation (8). This allows us to prevent steep discounts of $\gamma^n$ in $n$-step backup when using longer $n$-step returns. As a result, LNSS can effectively use large $N$ at a scale of 100 while D4PG (Barth-Maron et al., 2018) uses $n$ at a scale of 5 steps, and the same for Rainbow (Hessel et al., 2018).

2) LNSS can also be effective in a sparse reward environment. With a longer reward trajectory of $N$ future steps, LNSS continuously provides a reward estimate as feedback to the agent by assigning credits progressively backward from the time of achieving the desired states. That is to say that LNSS

has turned a sparse reward into a dense reward. Thus, this helps the agent effectively and efficiently learn the task. However, the single-step method does not have the ability to provide any guidance or feedback to the learning agent until reaching a goal.

3) The $Q$ value for policy $\pi$ due to LNSS differ from that in Equation 1. The former aims to maximize the sum of a sequence of surrogate rewards which are the weighted averages of the sequence of $N$-steps of the original rewards. As such, goal-related information can be progressively propagated back from the goal state to the current state. This prevents the agent from getting trapped from reaching its intended goal and thus being stuck in an undesirable set of states due to a lack of feedback.

## 5   Variance Analysis

We now analyze the behavior of an actor-critic RL and our LNSS actor-critic RL. We consider the infinite horizon discounted reward formulation of RL (with $0 < \gamma < 1$). Specifically, we show that the upper bound on the variance in $Q$ value due to LNSS differs by an exponential factor from that of a single step actor-critic RL (AC). As this upper bound reduces exponentially as $N$ increases, it suggests significant variance reduction by using LNSS from using single step reward.

We first represent the $Q$ values using single step reward $r_k$ in Equation (2) and using surrogate reward $r'_k$ from LNSS in Equation (8), respectively as follows,

$$\text{Var}\left[Q_{i+1}(s_k, a_k)\right] = \text{Var}\left[r_k\right] + \text{Var}\left[\gamma Q_i(s_{k+1}, a_{k+1})\right] + 2\text{Cov}\left[r_k, \gamma Q_i(s_{k+1}, a_{k+1})\right]. \tag{10}$$

$$\text{Var}\left[\mathbb{Q}_{i+1}(s_k, a_k)\right] = \text{Var}\left[r'_k\right] + \text{Var}\left[\gamma \mathbb{Q}_i(s_{k+1}, a_{k+1})\right] + 2\text{Cov}\left[r'_k, \gamma \mathbb{Q}_i(s_{k+1}, a_{k+1})\right]. \tag{11}$$

**Lemma 1**. Assume $\{r_k\}$ is IID and drawn from the memory buffer $\mathbb{D}$. Let $Q_i(s_{k+1}, a_{k+1})$ in Equation (8) be the $i$-th approximated return to solve Equation (1). We then have the following,

$$\text{Cov}\left[r_k, r_{j\neq k}\right] = 0, \tag{12}$$

$$\text{Cov}\left[r_k, Q_i(s_{k+1}, a_{k+1})\right] = 0. \tag{13}$$

**Theorem 1**. Consider the variances of two $Q$ value sequences, denoted as $Q_i$ and $\mathbb{Q}_i$, in Equation (10) and Equation (11), which are obtained respectively from a single step method and an LNSS method. Additionally, assume that $Q_0 = \text{Var}\left[Q_0\right] = 0$ and $\mathbb{Q}_0 = \text{Var}\left[\mathbb{Q}_0\right] = 0$. Let the IID reward $\{r_k\}$ and $\{r'_k\}$ be drawn from the memory buffer $\mathbb{D}$. Assume the variance of $\{r_k\}$ is upper bounded by a finite positive number $\mathbb{B}$, i.e., $\text{Var}\left[r_k\right] \leq \mathbb{B}$. Further define a constant $\psi$ as,

$$\psi = (\frac{\gamma - 1}{\gamma^N - 1})^2 (\frac{\gamma^{2N} - 1}{\gamma^2 - 1}). \tag{14}$$

Then the upper bounds of the variances of the two $Q$ value sequences, $\text{Var}\left[Q_{i+1}\right]$ and $\text{Var}\left[\mathbb{Q}_{i+1}\right]$, are respectively described below,

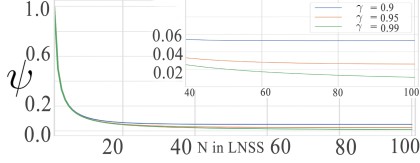

Figure 1: Variance discount factor $\psi$ in Equation (14).

$$\text{Var}\left[Q_{i+1}(s_k, a_k)\right] \leq \sum_{t=1}^{i+1} (\gamma^{t-1})^2 \mathbb{B}, \tag{15}$$

$$\text{Var}\left[\mathbb{Q}_{i+1}(s_k, a_k)\right] \leq \psi \sum_{t=1}^{i+1} (\gamma^{t-1})^2 \mathbb{B}. \tag{16}$$

**Proof.** Both proofs of Lemma 1 and Theorem 1 are given in Appendix A.

**Remark 2**. We now provide some insights based on the variance analysis above.
1) Given $\psi$ in Equation (14), i.e., $\psi = (\frac{\gamma - 1}{\gamma^N - 1})^2 (\frac{\gamma^{2N} - 1}{\gamma^2 - 1})$, it follows that for large $N$, $\psi = (\gamma - 1)^2 (\frac{-1}{\gamma^2 - 1}) = \frac{1 - \gamma}{1 + \gamma}$.
2) Furthermore, by the following identifies, $\gamma^2 - 1 = (\gamma - 1)(\gamma + 1)$ and $\gamma^{2N} - 1 = (\gamma^N - 1)(\gamma^N + 1)$, we have that $\psi = (\frac{\gamma - 1}{\gamma + 1})(1 + \frac{2}{\gamma^N - 1})$. Therefore, $\psi$ decreases exponentially (refer to Figure 1).
3) From inspecting Equations (15) and (16), we can see a clear advantage of using long "$N$-step" in LNSS over the typical reward $r_k$.

# 6 Experiments and Results

In this section, we first show how LNSS can benefit learning by using a simple Maze environment. To provide insight on the effect of using LNSS, we compare the performance of the original $Q$-learning with the one that the stage reward is replaced by LNSS reward.

We then provide a comprehensive evaluation of our proposed LNSS piggybacked on three promising DRL algorithms (DDPG, D4PG, TD3) by measuring their performance on several benchmarks in DMC and GYM (for results on GYM, please refer to Appendix E.5). Details of the implementation, training, and evaluation procedures are provided in Appendix B. In reporting evaluation results below, we use the following short-form descriptions.

1) "Base": the original DRL algorithms (TD3, D4PG, DDPG).

2) "LNSS": LNSS piggybacked on the respective DRL algorithms where $N$=100 unless otherwise specified.

3) "n5": the DRL algorithms with an "$n$-step" implementation as in Equation (3) with $n = 5$.

Our evaluations aim to quantitatively address the following questions:
Q1. How is LNSS different from original reward? and How it helps learning?
Q2. How does LNSS improve Base method?
Q3. How does LNSS compare with previous $n$-step method ($n5$)?
Q4. Is LNSS robust enough to compensate for stochastic reward?
Q5. Does LNSS improve Base methods under a sparse reward setting?
Q6. How does LNSS reduce the variance of estimated $Q$ values?
Q7. How different $N$ in LNSS affects performance?

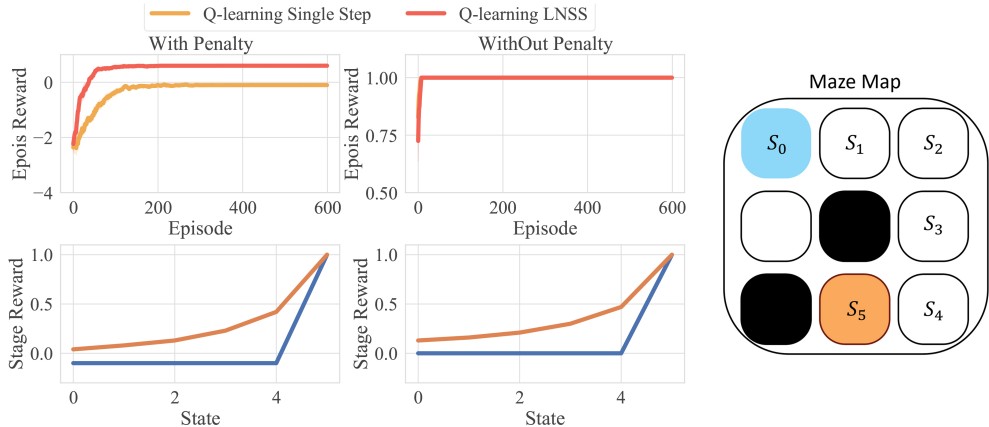

Figure 2: Evaluation of LNSS piggybacked on $Q$-learning in simple Maze (right panel). The evaluations are averaged over 10 seeds. The x-axis is the number of episodes (top-left panel) and the state number (bottom-left panel) in the plots for episode reward and stage reward, respectively. Note that the stage reward for LNSS is calculated based on Equation 6

## 6.1 The Simple Maze Problem

The right panel of Figure 2 shows the Maze environment where $S_0$ is the initial state, $S_5$ is the goal state, and black blocks are walls. The unique optimal policy is {right,right,down,down,left}. We consider two problem formulations:

1) Without Penalty: the agent only receives a reward 1 upon reaching goal state $S_5$. For all other states, the agent receives 0's.

2) With Penalty: the agent only receives a reward 1 upon reaching goal state $S_5$. For all other states, the agent receives -0.1 per stage as a penalty.

Q1 **LNSS takes future rewards into formulating the surrogate stage reward to guide a RL agent in reaching the goal state.** As Figure 2 lower-left panels show, the stage reward from LNSS is more

| No Penalty Case | | | | | | | | |
|---|---|---|---|---|---|---|---|---|
| | Up | | Down | | Right | | Left | |
| State | Original Reward | LNSS Reward | Original Reward | LNSS Reward | Original Reward | LNSS Reward | Original Reward | LNSS Reward |
| 0 | 0 | 0 | 0 | 0 | **0.47** | **1.33** | 0 | 0 |
| 1 | 0 | 0 | 0 | 0 | **0.62** | **1.45** | 0 | 0 |
| 2 | 0 | 0 | **0.76** | **1.48** | 0 | 0 | 0 | 0 |
| 3 | 0 | 0 | **0.89** | **1.36** | 0 | 0 | 0 | 0 |
| 4 | 0 | 0 | 0 | 0 | 0 | 0 | **0.99** | **0.99** |
| With Penalty Case | | | | | | | | |
| | Up | | Down | | Right | | Left | |
| State | Original Reward | LNSS Reward | Original Reward | LNSS Reward | Original Reward | LNSS Reward | Original Reward | LNSS Reward |
| 0 | **-0.09** | -0.05 | **-0.09** | -0.05 | **-0.09** | **1.1** | **-0.09** | -0.05 |
| 1 | -0.07 | -0.02 | **0** | -0.01 | -0.1 | **1.26** | -0.13 | -0.03 |
| 2 | -0.055 | 0 | -0.055 | **1.36** | **-0.054** | 0 | -0.055 | -0.01 |
| 3 | -0.03 | 0 | **0.04** | **1.3** | -0.03 | 0.03 | 0 | 0 |
| 4 | -0.01 | 0.03 | -0 | 0.06 | 0 | 0.07 | **0.38** | **0.997** |

Table 1: $Q$ tables of $Q$-learning using original reward and LNSS reward. The $Q$ values reported are averages across 10 different random seeds.

informative as LNSS provides sequences of gradual and smoother rewards to be used in $Q$-learning. As a result, from the upper-left panels, this informative guidance from LNSS significantly enhances the performance in terms of episode rewards. The effect is even more pronounced when dealing with penalties.

From Table 1, with the original reward, the $Q$ value is the sum of discounted expected rewards as in Equation 4. The farther away the goal state, the smaller the $Q$ values are as a result of discounts over the horizon. When learning under the environment of using penalty rewards, the $Q$ values in States $S_0, S_1, S_2$ show that they are even farther away from the goal state value. The $Q$ values can not be effectively updated due to a lack of informative feedback signal, and thus, causing the agent to get stuck and cease learning. In comparison, with the LNSS reward, goal-related information can be propagated back from the goal state to the initial state. This guidance plays a crucial role in assisting agents to overcome situations where they might otherwise get stuck.

| | Acrobot SwingUp | | | Humanoid Walk | | | Fish Swim | | | Finger TurnHard | | |
|---|---|---|---|---|---|---|---|---|---|---|---|---|
| Algorithm | Success [%] | Avg. Rwd $[\mu \pm 2\sigma]$ | Rank [%] | Success [%] | Avg. Rwd $[\mu \pm 2\sigma]$ | Rank [%] | Success [%] | Avg. Rwd $[\mu \pm 2\sigma]$ | Rank [%] | Success [%] | Avg. Rwd $[\mu \pm 2\sigma]$ | Rank [%] |
| Noise Level 0% (0.0) | | | | | | | | | | | | |
| **DDPG-LNSS** | **100** | **304.6 ± 82.9** | **-25.1** | **100** | **250.9 ± 16.5** | **-34.6** | **100** | **645.1 ± 86.9** | **-12.7** | **100** | **943.3 ± 32.9** | **0** |
| DDPG-Base | 100 | 187.9 ± 68.5 | -53.8 | 0 | 1.3 ± 0.4 | -99.6 | 100 | 272.9 ± 27.3 | -63.1 | 80 | 301.8 ± 160.3 | -68.0 |
| DDPG-n5 | 100 | 195.2 ± 67.6 | -51.9 | 0 | 2.2 ± 1.5 | -99.4 | 100 | 150.5 ± 33.7 | -79.6 | 100 | 459.8 ± 126.3 | -51.3 |
| **TD3-LNSS** | **100** | **114.8 ± 33.8** | **-71.8** | **100** | **307.9 ± 16.9** | **-19.8** | **100** | **676.8 ± 44.2** | **-8.4** | **100** | **940.1 ± 29.6** | **-0.3** |
| TD3-Base | 0 | 5.2 ± 4.1 | -98.7 | 60 | 184.2 ± 178.8 | -52.1 | 100 | 483.1 ± 127.6 | -34.6 | 90 | 251.8 ± 101.4 | -73.3 |
| TD3-n5 | 70 | 61.7 ± 45.7 | -84.8 | 0 | 1.01 ± 0.41 | -99.6 | 100 | 731.7 ± 132.3 | -1.1 | 90 | 394.2 ± 164.4 | -58.2 |
| **D4PG-LNSS** | **100** | **406.3 ± 13.1** | **0** | **100** | **383.9 ± 37.1** | **0** | **100** | **738.7 ± 59.8** | **0** | **100** | **811.5 ± 59.9** | **-14.0** |
| D4PG-Base | 100 | 133.5 ± 16.2 | -67.1 | 100 | 277.7 ± 79.8 | -27.7 | 100 | 585.7 ± 36.1 | -20.7 | 100 | 336.9 ± 107.8 | -64.3 |
| D4PG-n5 | 100 | 310.1 ± 49.5 | -23.7 | 100 | 360.3 ± 50.4 | -6.2 | 100 | 683.5 ± 95.6 | -7.5 | 90 | 413.7 ± 243.8 | -56.1 |
| Noise Level 1% (0.01) | | | | | | | | | | | | |
| **DDPG-LNSS** | **100** | **243.5 ± 62.3** | **-40.1** | **100** | **255.1 ± 22.6** | **-33.6** | **100** | **700.4 ± 107.6** | **-5.2** | **100** | **934.3 ± 30.58** | **-0.1** |
| DDPG-Base | 100 | 188.2 ± 58.7 | -53.7 | 0 | 0.99 ± 0.25 | -99.8 | 100 | 340.1 ± 135.4 | -53.9 | 100 | 302.5 ± 130.4 | -67.9 |
| DDPG-n5 | 100 | 159.6 ± 25.8 | -60.7 | 0 | 1.15 ± 0.43 | -99.7 | 100 | 149.4 ± 59.1 | -55.8 | 100 | 416.9 ± 168.4 | -55.8 |
| **TD3-LNSS** | **100** | **109.3 ± 20.8** | **-73.1** | **100** | **197.6 ± 45.1** | **-48.5** | **100** | **700.1 ± 84.5** | **-5.2** | **100** | **912.8 ± 92.2** | **-3.2** |
| TD3-Base | 0 | 2.4 ± 1.53 | -99.4 | 0 | 2.05 ± 1.43 | -99.5 | 100 | 506.9 ± 111.5 | -31.4 | 60 | 193.4 ± 142.2 | -79.5 |
| TD3-n5 | 70 | 69.6 ± 23.5 | -82.9 | 0 | 1.55 ± 0.8 | -99.6 | 100 | 688.7 ± 132.3 | -6.8 | 90 | 384.1 ± 232.4 | -59.3 |
| **D4PG-LNSS** | **100** | **330.3 ± 60.6** | **-18.7** | **100** | **247.9 ± 42.3** | **-35.4** | **100** | **688.3 ± 55.3** | **-6.8** | **100** | **801.9 ± 80.2** | **-15.0** |
| D4PG-Base | 100 | 136.5 ± 30.2 | -66.4 | 100 | 210.5 ± 44 | -45.2 | 100 | 650.2 ± 89.3 | -11.9 | 100 | 300.5 ± 143.2 | -68.1 |
| D4PG-n5 | 100 | 262.9 ± 67.1 | -35.3 | 0 | 10.9 ± 2.3 | -97.2 | 100 | 578.7 ± 78.9 | -21.7 | 90 | 383.7 ± 293.8 | -59.3 |
| Noise Level 10% (0.1) | | | | | | | | | | | | |
| **DDPG-LNSS** | **100** | **232.6 ± 61.5** | **-42.8** | **100** | **235.4 ± 25.7** | **-38.7** | **100** | **699.5 ± 95.3** | **-5.3** | **100** | **940.2 ± 54.1** | **-0.3** |
| DDPG-Base | 100 | 170.7 ± 42.1 | -58.0 | 0 | 0.76 ± 0.42 | -99.8 | 100 | 330.42 ± 144.2 | -55.3 | 100 | 315.2 ± 205.4 | -66.6 |
| DDPG-n5 | 100 | 139.1 ± 28.7 | -65.8 | 0 | 1.3 ± 0.9 | -99.7 | 100 | 116.5 ± 16.8 | -84.2 | 100 | 456.5 ± 219.7 | -51.6 |
| **TD3-LNSS** | **100** | **80.7 ± 15.73** | **-80.1** | **100** | **200.2 ± 117.6** | **-47.3** | **100** | **696.5 ± 174.1** | **-5.7** | **100** | **888.54 ± 92** | **-5.8** |
| TD3-Base | 0 | 2.19 ± 1.35 | -99.5 | 0 | 1.5 ± 0.33 | -99.7 | 100 | 454.9 ± 272.9 | -38.4 | 60 | 190.1 ± 215.7 | -79.9 |
| TD3-n5 | 0 | 24.9 ± 22.7 | -93.9 | 0 | 1.1 ± 0.28 | -99.7 | 100 | 654.7 ± 130.2 | -11.4 | 80 | 338.1 ± 216.4 | -64.2 |
| **D4PG-LNSS** | **100** | **331.3 ± 49.5** | **-18.5** | **100** | **240.8 ± 46.3** | **-37.3** | **100** | **681.4 ± 212.6** | **-7.8** | **100** | **803.9 ± 83.3** | **-14.8** |
| D4PG-Base | 100 | 130.1 ± 28.9 | -67.9 | 0 | 1.42 ± 0.7 | -99.7 | 100 | 645.3 ± 245.2 | -12.6 | 100 | 302.5 ± 153.2 | -67.9 |
| D4PG-n5 | 100 | 251.4 ± 56.4 | -38.1 | 0 | 0.89 ± 0.6 | -99.8 | 100 | 563.4 ± 214.7 | -23.7 | 90 | 355.1 ± 285.5 | -62.4 |

Table 2: Systematic evaluations of LNSS respectively augmented Base algorithms, and comparisons to the Base and $n5$-augmented Base algorithms. "Rank" (%) is the "percent of reward difference", the closer it is to 0 the better.

## 6.2 Main Results

Evaluation results in Table 2 are performed using default setup of Acrobot SwingUp, Humanoid Walk, Fish Swim, and Finger TurnHard in DMC. In the Table 2, "Success" is shorthand for success rate, "Avg. Rwd" stands for average reward, and "Rank" (%) is the "percent of reward difference",

which is ( the average reward of the evaluated algorithm over that of the top performing algorithm - 1), the closer it is to 0 the better. Note that, in computing the success rate, only those trials that have achieved a reward of at least 50 are accounted for as successes.

The results are obtained at different noise levels, based on the last 50 evaluations of 10 different random seeds (same for all compared algorithms). Best performances are boldfaced for average reward (Avg. Rwd).

**Q2 LNSS improves Base methods.** The learning curves for the four continuous dense reward benchmark environments are shown in Figure 3. Quantitative comparisons to the baselines are summarized in the first section of Table 2 (noise = 0). Overall, LNSS (solid lines) outperforms their respective base methods (dash lines) in terms of averaged reward (Awg.Rwd), learning speed, and success rate for all three DRL algorithms. Among the measures, the success (learning) rate is to address the random initialization challenge caused by random seeds [10]. LNSS method helps its Base method to achieve a 100% learning success rates whereas DDPG and TD3 Base method struggle with Humanoid Walk and Finger TurnHard. Besides, according to the "Rank" measure in Table 2, LNSS helps enhance the performances of almost all Base methods up to the top-performing algorithm in each category in all the tested environments. Detailed plots of pair-wise comparison for without noise condition are provided in appendix E.1.

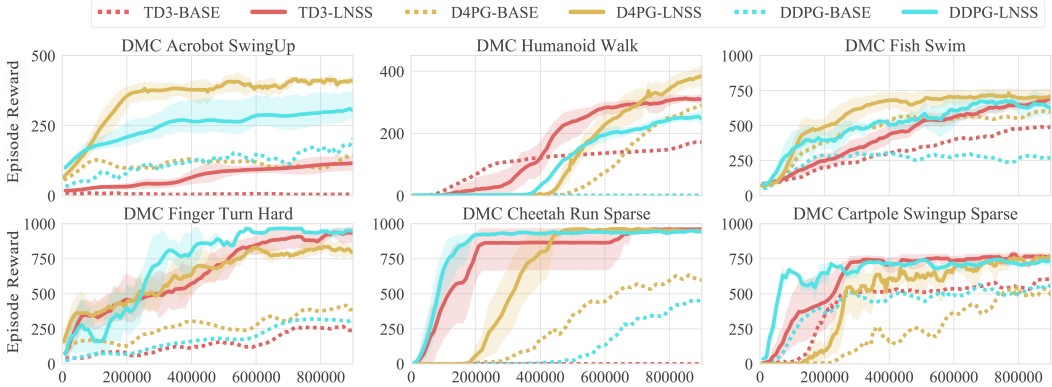

Figure 3: Systematic evaluation of LNSS piggybacked on three DRL algorithms (DDPG, D4PG, TD3) in DMC environments without reward noise. The shaded regions represent the 95 % confidence range of the evaluations over 10 seeds. The x-axis is the number of steps. Detailed pair-wise comparison plots are provided in appendix E.1.

**Q3 LNSS outperforms previous "$n$-step" methods ($n5$).** The main improvement of LNSS over $n5$ are reflected in improved success rate, averaged reward, and "Rank". This improvement can be quantitatively viewed in the first section of Table 2 (noise = 0). LNSS consistently enables the Base methods to achieve 100% learning success rate. However, the $n5$ method may even cause some learning to fail such as TD3 in Humanoid Walk and D4PG in Finger TurnHard. Additionally, according to "Rank" in Table 2, LNSS methods enable all cases to reach performances near those from the best algorithms. However, only TD3-$n5$ has resulted in slightly better "Rank" in Fish Swim only for without noise condition while all other $n5$ methods are below LNSS or even below its Base such as TD3-$n5$ for humanoid walk and DDPG-$n5$ for Fish swim. We also observe that LNSS has improved the average reward and lowered the standard deviation for all Base algorithms over those by the $n5$ method.

**Q4 LNSS is robust under noisy rewards.** We performed the same set of evaluations in Table 2 in DMC experiments under random uniform reward noise settings with magnitudes at 1% and 10% of the maximum DMC step reward (1). As results in the appendix E.2 (noise level 1%), Appendix E.3 (Noise level 10%), and the last two sections in Table 2 (noise = 1% and 10%) show, at larger noise level, all algorithms have degraded performance with lower mean averaged reward, larger standard division, and more negative "Rank". However, LNSS consistently outperformed Base and $n5$ methods in terms of all measures for all three DRL algorithms. It is important to point out that we observe LNSS methods robustly improve the success rate with 100% for all environments. However, we observe that both Base and $n5$ methods suffer greatly due to the increased noise level such as

in Humanoid Walk and Finger Turn Harde environments. As Equation (3) shows the performance degradation for $n5$ methods are due to accumulated noise when collecting the discounted sum of returns. This accumulated noise will induce a much larger approximation variance and result in degraded performance.

**Q**5 **LNSS is effective in environments with sparse rewards.** For setting up sparse rewards in Cheetah environments, refer to appendix B. As Cheetah Sparse and Cartpole Sparse result in Figure 3. One can observe that LNSS enables all DRL algorithms successfully learn all three sparse reward tasks with converged episode rewards greater than 800. However, due to the limited feedback from sparse rewards, all base methods struggle to learn and even fail to learn (e.g, Cheetah Run). Moreover, it is important to point out that LNSS helps improve the learning speed over the base methods. This is expected as a long future reward trajectory of $N$-step is able to provide continuous reward feedback to the agent by assigning credits progressively backward from the time of achieving the desired states. In contrast, a single-step method does not provide any guidance or feedback to the learning agent until reaching the goal.

**Q**6 **LNSS helps reduce the variance of $Q$ value estimation**. For this evaluation, we let coefficient of variation of $Q$ value (in percentage) as $cv(Q) = \frac{std(Q)}{Q}\%$. In Figure 4, we show $cv(Q)$ every 2e5 steps where as an example, we show TD3-LNSS outperforms TD3-Base and TD3-n5 under 1% noise in rewards. Similar results are obtained for DDPG and D4PG classes of algorithms and shown in appendix E.4. Also shown in the same Appendix are respective results at different noise levels. Results (Figure 12 to Figure 20 in appendix E.4) also show that for each LNSS augmented DRL algorithms, it has resulted in improved performance in $cv(Q)$ to near the best performance among all DRL algorithms. Figure 4 provides an empirical validation for our theoretical result on bounds of variances of the $Q$ value, namely, with large LNSS parameter $N$ or at the late stage of learning, $Q$ value variances become smaller as the upper bound of $Q$ value variance decreases significantly according to Equation (14) in Theorem 1.

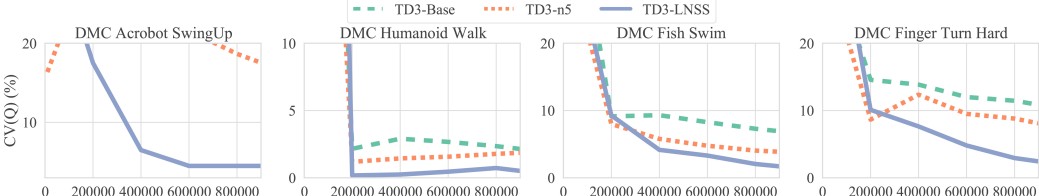

Figure 4: The $cv(Q)$ (coefficient of variaiton of $Q$ value, the lower the better) of TD3 in DMC with 1% noise. Results are averaged over 10 seeds. The x-axis is the number of steps. Additional and respective results for DDPG and D4PG can be found in appendix E.4

**Q**7. **Effects of different length $N$ in LNSS.** In Figure 5, we show the effect of different choices of $N$ ($N = 5, 10, 20, 50, 100$) in our proposed LNSS on top of TD3. (Similar results hold for D4PG and DDPG which can be found in appendix E.7.) Comparisons are performed using the same set of hyperparameters except a varying $N$. Figure 5 depicts performance curves for DMC tasks. The LNSS with $N = 50$ and $N = 100$ outperform LNSS with $N = 5, 10$ in terms of final reward and learning speed. The results corroborate what we expect from our theoretical analysis (please refer to Figure 1 in the paper). We can see a clear benefit of using large $N$ ($N = 50$, and 100 ) over small($N = 5$, and 10).

Additionally, in Figure 6, we show how $cv(Q)$ (coefficient of variation of $Q$) varies with different $N$. Among the three $N$ values, $N = 100$ outperforms others. Also, we observe that, in more complex environment, a longer $N$ is favorable. Notice that, for cartpole sparse environment, there is little difference among different $N$ values. However, in Finger TurnHard and Humanoid Walk, the $cv(Q)$ values of Base cases and $N = 5$ cases increase dramatically. However, the $cv(Q)$ values for $N = 50$ and $N = 100$ change only slightly. Figure 6 provides an empirical validation for our theoretical analysis on variance discount factor $\psi$ in Figure 1 that the longer the $N$, the more reduction in variance of $Q$ value estimation.

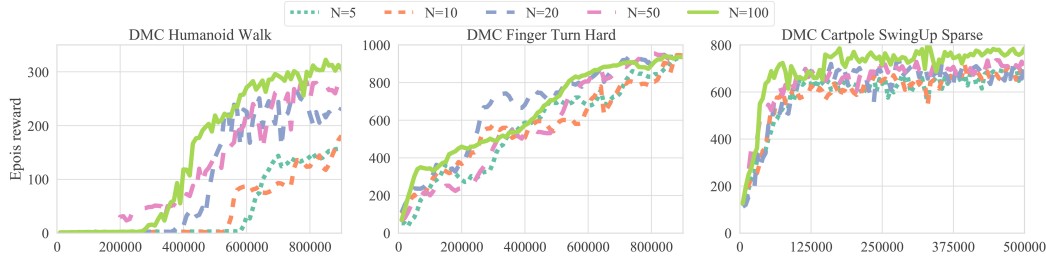

Figure 5: Episode rewards for the 3 tasks in DMC by TD3-LNSS with different $N$ ($N = 5, 10, 20, 50, 100$) in Equation (6). The results are averaged from 10 seeds. The x-axis is the number of steps. Respective results for DDPG and D4PG are provided in appendix E.7

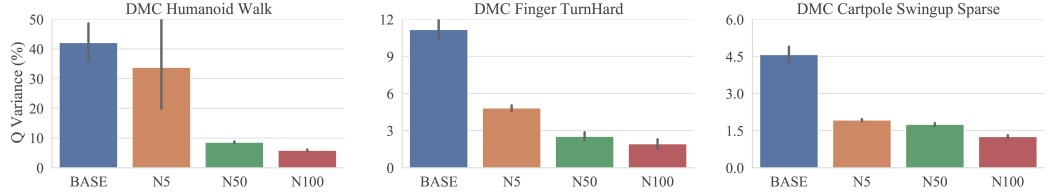

Figure 6: $cv(Q)$ (coefficient of variaiton of $Q$ value, the lower the better) for the 3 tasks in DMC by LNSS with different $N$ ($N = 5, 50, 100$) in Equation (6).

### 6.3 Limitation of This Study.

Here we introduce a new multi-step method, the LNSS, to effectively utilize longer reward trajectories of future steps than those used in existing methods to estimate the $Q$ value with an aim of reducing variances in learning. By piggybacking LNSS on top of TD3, DDPG, and D4PG, we demonstrate improved performance in terms of final reward, learning speed, and variance reduction in $Q$ value. However, there exist two limitations. 1) The LNSS requires positive semi-definite reward values in a long trajectory. Large negative reward may wash out the significance of good positive rewards. However, this limitation can be easily fixed by elevated reward stored in the temporary buffer $\mathbb{D}'$ by a positive constant, and lower bounded it by 0. 2) Early termination is another limitation during implementation, which mostly affects LNSS in GYM environments (Hopper, Humanoid, Walker2d). Different from DMC tasks, GYM allows an environment to terminate before reaching the maximum time steps. As such, in early training stage, only 10 or 20 steps may be used to compute LNSS which diminishes the power of large $N$ (such as 50 or 100). To resolve this limitation, re-programming task settings in the original GYM environments is required.

## 7 Discussion and Conclusion

1) In this work, we introduce a novel "$N$-step" method, the LNSS which utilize longer reward trajectories of future steps than those used in existing methods to estimate the $Q$ value with an aim of reducing variances in learning. It is easy to implement, as shown in the paper, and it can be easily piggybacked on policy gradient DRL algorithms. It has been shown consistently outperform respective Base and short $n$-step algorithms in solving benchmark tasks in terms of performance score, learning success rate, and convergence speed. 2) We provide a theoretical analysis to show that LNSS reduces the upper bound of variance in $Q$ value exponentially from respective single step methods. 3) We empirically demonstrate the performance of LNSS piggybacked on TD3, DDPG, and D4PG in a variety of benchmark environments that have been challenging for existing methods to obtain good results. 4) We show LNSS can provide consistent performance improvement under various reward noise settings, and for sparse rewards. Our results suggest that LNSS is a promising tool for improving learning speed, learning performance score, and reducing learning variance. Further investigation on how to maximize the benefit of selecting an optimal reward length $N$, and how to take advantage of a different $n$ in $Q$ value update are exciting questions to be addressed in future work.

## 8  Acknowledgment

This research was supported in part under NSF grants #1808752 and #2211740.

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
