**Proof**. Given that $\{r_k\}$ is IID, we reach Equation (19) immediately. Based on Equation (1), the conditional probability distribution of $Q_i(s_{k+1}, a_{k+1})$ only depends on state action pair $(s_{k+1}, a_{k+1})$. It thus leads to Equation (20).

Based on Lemma 1, we now analyze the variance of the $Q$ value sequence.

**Theorem 1**. Consider the variances of two $Q$ value sequences, denoted as $Q_i$ and $\mathbb{Q}_i$, in Equation (17) and Equation (18), which are obtained respectively from a single step method and an LNSS method. Additionally, assume that $Q_0 = \text{Var}\left[Q_0\right] = 0$ and $\mathbb{Q}_0 = \text{Var}\left[\mathbb{Q}_0\right] = 0$. Let the IID reward $\{r_k\}$ and $\{r'_k\}$ be drawn from the memory buffer $\mathbb{D}$. Assume the variance of $\{r_k\}$ is upper bounded by a finite positive number $\mathbb{B}$, i.e., $\text{Var}\left[r_k\right] \leq \mathbb{B}$. Further define a constant $\psi$ as,

$$\psi = \left(\frac{\gamma - 1}{\gamma^N - 1}\right)^2 \left(\frac{\gamma^{2N} - 1}{\gamma^2 - 1}\right). \quad (21)$$

Then the upper bounds of the variances of the two $Q$ value sequences, $\text{Var}\left[Q_{i+1}\right]$ and $\text{Var}\left[\mathbb{Q}_{i+1}\right]$, are respectively described below,

$$\text{Var}\left[Q_{i+1}(s_k, a_k)\right] \leq \sum_{t=1}^{i+1} (\gamma^{t-1})^2 \mathbb{B}, \quad (22)$$

$$\text{Var}\left[\mathbb{Q}_{i+1}(s_k, a_k)\right] \leq \psi \sum_{t=1}^{i+1} (\gamma^{t-1})^2 \mathbb{B}. \quad (23)$$

**Proof.** We prove by mathematical induction. First let $i = 0$. Based on Equation (20), Equations (17) and (18) become:

$$\begin{aligned} \text{Var}\left[Q_1(s_k, a_k)\right] &= \text{Var}\left[r_k\right], \\ \text{Var}\left[\mathbb{Q}_1(s_k, a_k)\right] &= \text{Var}\left[r'_k\right]. \end{aligned} \quad (24)$$

First note that $\text{Var}\left[Q_1(s_k, a_k)\right] \leq \mathbb{B}$ as $\text{Var}\left[r_k\right] \leq \mathbb{B}$.

Then from Equation (6), $\mathrm{Var}\,[r'_k]$ can be written as:

$$
\begin{aligned}
\mathrm{Var}\,[r'_k] &= \mathrm{Var}\,\big[\frac{\gamma-1}{\gamma^N-1}\sum_{t=k}^{k+N-1}\gamma^{t-k}r_t\big] \\
&= (\frac{\gamma-1}{\gamma^N-1})^2\mathrm{Var}\,\big[\sum_{t=k}^{k+N-1}\gamma^{t-k}r_t\big] \\
&= (\frac{\gamma-1}{\gamma^N-1})^2\sum_{l=k}^{k+N-1}\sum_{t=k}^{k+N-1}(\gamma^{t-k})^2\mathrm{Cov}\,(r_t,r_l) \\
&= (\frac{\gamma-1}{\gamma^N-1})^2\sum_{t=k}^{k+N-1}(\gamma^{t-k})^2\mathrm{Var}\,[r_t] \\
&\le (\frac{\gamma-1}{\gamma^N-1})^2(\frac{\gamma^{2N}-1}{\gamma^2-1})\mathbb{B},
\end{aligned}
\tag{25}
$$

where in the above, the equation above the inequality is obtained from applying Lemma 1.

We thus have,

$$
\mathrm{Var}\,[\mathbb{Q}_1(s_k,a_k)] \le \psi\mathbb{B}.
\tag{26}
$$

This shows that the theorem holds for $i=0$.

Then Assume that Equation (22) and (23) hold for $i=l-1$, where $l=1,2,....$ Then, for $i=l$, we have

$$
\begin{aligned}
\mathrm{Var}\,[Q_{l+1}(s_k,a_k)] &= \mathrm{Var}\,[r_k] + \mathrm{Var}\,[\gamma Q_l(s_{k+1},a_{k+1})] \\
&\le \mathbb{B} + \gamma^2\sum_{t=1}^{l}(\gamma^{t-1})^2\mathbb{B} \\
&= \sum_{t=1}^{l+1}(\gamma^{t-1})^2\mathbb{B}.
\end{aligned}
\tag{27}
$$

Additionally,

$$
\begin{aligned}
\mathrm{Var}\,[\mathbb{Q}_{l+1}(s_k,a_k)] &= \mathrm{Var}\,[r'_k] + \mathrm{Var}\,[\gamma\mathbb{Q}_i(s_{k+1},a_{k+1})] \\
&\le \psi\mathbb{B} + \gamma^2\psi\sum_{t=1}^{l}(\gamma^{t-1})^2\mathbb{B} \\
&= \psi\sum_{t=1}^{l+1}(\gamma^{t-1})^2\mathbb{B}.
\end{aligned}
\tag{28}
$$

Thus Theorem 1 holds.

## Appendix B    Additional Implementation Details

We use PyTorch for all implementations. All results were obtained using our internal server consisting of AMD Ryzen Threadripper 3970X Processor, a desktop with Intel Core i7-9700K processor and a desktop with AMD Ryzen 7 3800XT processor. Our implementation code is detailed and provided in Github link and Appendix. C

**Training Procedure**.

An episode is initialized by resetting the environment, and terminated at max step $T=1000$ or early termination if criteria are met depending on specific tasks. A trial is a complete training process contains a series of consecutive episodes. Each trial is run for a maximum $9\times10^5$ time steps with evaluations at every $1\times10^4$ time steps. Each task is reported over 10 trials where the environment and the network were initialized by 10 mother random seeds, $(0-9)$ in this study.

For each training trial, to remove the dependency on the initial parameters of a policy, we use a purely exploratory policy for the first 8000 time steps (start timesteps). Afterwards, we use an off-policy exploration strategy, adding Gaussian noise $\mathcal{N}(0,0.1)$ to each action.

**Evaluation Procedure**.

Every $1 \times 10^4$ time steps training, we have an evaluation section and each evaluation reports the average reward over 5 evaluation episodes, with no exploration noise and with fixed policy weights. The random seeds for evaluation are different from those in training which are shown in the following section on Random Seed.

**Random Seed**.

Each of the tasks we used in this work was trained for 10 trials with 10 different mother random seeds $s_m$ (0,1,...,9) across all algorithms. Within each trial, evaluations were performed using seeds $(s_m + 100)$. For distributed training, we implement $n$ parallel actors to generate experiences and each actor $a_i, i = 0, 1, 2, ...n$ has a seed $s_{a_i}$ defined as $s_{a_i} = s_m + i$. The seeds $s_{a_i}$ are used in the environment, Numpy, and PyTorch for seed initialization.

**Network Structure and optimizer**.

**TD3**.The actor-critic networks in TD3 are implemented by feedforward neural networks with three layers of weights. Each layer has 256 hidden nodes with rectified linear units (ReLU) for both the actor and critic. The input layer of actor has the same dimension as observation state. The output layer of the actor has the same dimension as action requirement with a tanh unit. Critic receives both state and action as input to THE first layer and the output layer of critic has 1 linear unit to produce $Q$ value. Network parameters are updated using Adam optimizer with a learning rate of $10^{-3}$ for simple control problems and for harder problems, we set the learning rate to a smaller value of $5 * 10^{-5}$. After each time step $k$, the networks are trained with a mini-batch of a 256 transitions $(s, a, r, s')$, $(s, a, r', s')$ in case of LNSS, sampled uniformly from a replay buffer $\mathbb{D}$ containing the entire history of the agent.

**D4PG**. Same with the actor-critic networks in D4PG are implemented by feedforward neural networks with three layers of weights. Each layer has 256 hidden nodes with rectified linear units (ReLU) for both the actor and critic. The input layer of actor has the same dimension as observation state. The output layer of the actor has the same dimension as action requirement with a tanh unit. Critic receives both state and action as input to THE first layer and the output layer of critic has a distribution with hyperparameters for the number of atoms $l$, and the bounds on the support $(V_{min}, V_{max})$. Network parameters are updated using Adam optimizer with a learning rate of $10^{-3}$. After each time step $k$, the networks are trained with a mini-batch of 256 transitions $(s, a, r, s')$, $(s, a, r', s')$ in case of LNSS, sampled uniformly from a replay buffer $\mathbb{D}$ containing the entire history of the agent.

**DDPG**. The actor-critic networks in DDPG are implemented by feedforward neural networks with three layers of weights. Each layer has 256 hidden nodes with rectified linear units (ReLU) for both the actor and critic. The input layer of actor has the same dimension as observation state. The output layer of the actor has the same dimension as action requirement with a tanh unit. Critic receives both state and action as input to the first layer and the output layer of critic has 1 linear unit to produce $Q$ value. Network parameters are updated using Adam optimizer with a learning rate of $10^{-3}$ for simple control problems and for harder problems, we set the learning rate to a smaller value of $5 * 10^{-5}$. After each time step $k$, the networks are trained with a mini-batch of a 256 transitions $(s, a, r, s')$, $(s, a, r', s')$ in case of LNSS, sampled uniformly from a replay buffer $\mathbb{D}$ containing the entire history of the agent.

**Hyperparameters**.

For TD3, target policy smoothing is implemented by adding $\epsilon \sim \mathcal{N}(0, 0.2)$ to the actions chosen by the target actor-network, clipped to $(-0.5, 0.5)$, delayed policy updates consist of only updating the actor and target critic network every $d$ iterations, with $d = 2$. While a larger $d$ would result in a larger benefit with respect to accumulating errors, for fair comparison, the critics are only trained once per time step, and training the actor for too few iterations would cripple learning. Both target networks are updated with $\tau = 0.005$.

The TD3 used in this study is based on the paper [7] and the code from the authors (https://github.com/sfujim/TD3). The distributed learning process is based on [34].

| Hyperparameter TD3 | Value |
|---|---|
| Start timesteps | 8000 steps |
| Evaluation frequency | 10000 steps |
| Max timesteps | 9e5 steps |
| Exploration noise | $\mathcal{N}(0, 0.1)$ |
| Policy noise | $\mathcal{N}(0, 0.2)$ |
| Noise clip | $\pm 0.5$ |
| Policy update frequency | 2 |
| Batch size | 256 |
| Buffer size | 1e6 |
| $\gamma$ | 0.99 |
| $\tau$ | 0.005 |
| Number of parallel actor | 8 (20 ) |
| LNSS-N | choose as results shows |
| Adam Learning rate | 1e-3 (5e-5) |

Table 3: TD3 + LNSS hyper parameters used for the GYM and DMC benckmark tasks

The D4PG used in this study is based on paper [6] and the code is modified from TD3 using the same distributed learning process as TD3.

| Hyperparameter D4PG | Value |
|---|---|
| Start timesteps | 8000 steps |
| Evaluation frequency | 10000 steps |
| Max timesteps | 9e5 steps |
| Exploration noise | $\mathcal{N}(0, 0.1)$ |
| Noise clip | $\pm 0.5$ |
| Batch size | 256 |
| Buffer size | 1e6 |
| $\gamma$ | 0.99 |
| $\tau$ | 0.005 |
| Number of parallel actor | 8 |
| LNSS-N | choose as results shows |
| Adam Learning rate | 1e-3 |
| $V_{max}$ | 100 |
| $V_{min}$ | 0 |
| $l$ | 51 |

Table 4: D4PG + LNSS hyper parameters used for the DMC benckmark tasks

The DDPG used in this study is based on paper [3] and the code is from the paper and the hyperparameter is from Table 5.

**Distributed Learning Procedure**. Distributed Learning is widely used in DRL algorithms [4, 6, 20] for speeding up experience gathering. Note from Equation (30) and (31) that updating the actor and the critic relies on sampling from some state distributions $p_\pi(s)$. We can parallelize this process by using a distributed process of multiple independent actors, each writing to the same memory buffer. Samples of experiences from the memory buffer can then be used in learning.

To speed up the learning process, we use a distributed implementation to parallelize computation. In the style of [34], we use a centralized agent with several workers operating in parallel. Each worker loads the most recent policy, interacts with the environment, and sends its observations to the central agent. Given the computing resource, 8 workers for simple control problems and 20 workers for harder problems were implemented in this study. All algorithm hyper-parameters are summarized in Table 3, Table 4, and Table 5.

**Sparse Reward Setup**. 1) Cheetah Run Sparse: Cheetah needs to run forward as fast as possible. The agent gets a reward only after speed exceeds 2.8 $m/s$, making the reward sparse. $r = 1$. That is,

| Hyperparameter DDPG | Value |
| --- | --- |
| Start timesteps | 8000 steps |
| Evaluation frequency | 10000 steps |
| Max timesteps | 9e5 steps |
| Exploration noise | $\mathcal{N}(0, 0.1)$ |
| Policy noise | $\mathcal{N}(0, 0.2)$ |
| Noise clip | $\pm 0.5$ |
| Policy update frequency | 2 |
| Batch size | 256 |
| Buffer size | 1e6 |
| $\gamma$ | 0.99 |
| $\tau$ | 0.005 |
| Number of parallel actor | 8 (20 ) |
| LNSS-N | choose as results shows |
| Adam Learning rate | 1e-3 (5e-5) |

Table 5: DDPG + LNSS hyper parameters used for the DMC benckmark tasks

if $v >= 2.8$ else $r = 0$.

# Appendix C   Code Details

Our implementation code is detailed and provided in Github link

---

**Algorithm 1** Long $N$-step Surrogate Stage (LNSS) Reward

---

**Given**:

- an RL algorithm $\mathbb{A}$, e.g DDPG,TD3,D4PG
- "$N$-step" number $N$
- $n$-step update $n$
- an experience buffer $\mathbb{D}$
- a temporary experience buffer $\mathbb{D}'$ with size $N$
- Total training episode $\mathbb{T}$

**Initialize**: $\mathbb{A}, \mathbb{D}, \mathbb{D}'$

1: **for** episode = 1, $\mathbb{T}$ **do**
2:      Reset initialize state $s_0$, $\mathbb{D}'$
3:      **for** k = 0, $T$ **do**
4:          Choose an action $a_k$ based on current state $s_k$ and learned policy from $\mathbb{A}$.
5:          Execute the action $a_k$ and observe a new state $s_{k+1}$ with reward signal $r_k$
6:          Store the transition $(s_k, a_k, r_k, s_{k+1})$ in $\mathbb{D}'$
7:          **if** $k + N - 1 \leq T$ **then**
8:             Get earliest memory $(s'_0, a'_0, r'_0, s'_1)$ in the $\mathbb{D}'$
9:             Calculate $r'$ based on Equation (7)
10:         Store the transition $(s'_0, a'_0, r', s'_1)$ in $\mathbb{D}$
11:         Clear transition $(s'_0, a'_0, r'_0, s'_1)$ in the $\mathbb{D}'$
12:         **else**
13:             Repeat step 8 to 11 and Calculate $r'$ based on Equation

$$r'_k = \frac{\gamma - 1}{\gamma^{T-k+1} - 1} \sum_{t=k}^{T} \gamma^{t-k} r_t. \tag{29}$$

14:         **end if**
15:         using $r'$ to perform $n$-step of optimization using $\mathbb{A}$ and mini-batch data from $\mathbb{D}$
16:      **end for**
17: **end for**

---

## Appendix D  Background

Here we present some DRL algorithm details regarding signle step updates and $n$-step updates related to our work:

1) DDPG [3] is a well-established off-policy policy gradient method. As an actor-critic (AC) algorithm, it contains two steps of policy evaluation (computing value function for a policy) and policy improvement (using value function to find a better policy). The policy ($\pi_\phi$) is called an actor and the action value function ($Q_\theta(s_k, a_k)$) is call a critic where both the actor and the critic are estimated by deep neural networks with parameters $\phi$ and $\theta$ respectively. Most AC methods are based on Bellman equation. The target equation becomes $y_k = r_k + \gamma Q_\theta(s_{k+1}, a_{k+1})$ so that the critic value $Q_\theta$ is updated by minimizing the loss function with respect to the weights ($\theta$) as:

$$L(\theta) = \mathbb{E}_{s \sim p_\pi, a \sim \pi}[(y - Q_\theta(s_k, a_k))^2]. \tag{30}$$

The actor weights can be updated by applying chain rule return from the start distribution $J$ with respect to the policy parameter $\phi$ :

$$\nabla_\phi J(\phi) = \mathbb{E}_{s \sim p_{\pi_\phi}} \left[ \nabla_a Q_\theta(s_k, a_k)|_{a_k = \pi_\phi(s_k)} \nabla_\phi \pi_\phi(s_k) \right]. \tag{31}$$

2) TD3 [7] is based on DDPG but uses a clipped double $Q$ network idea, where $Q_{\theta_j}(s_k, a_k)$ ($j = 1, 2$) represents the two respective $Q$ values. It takes the lesser value between the two, thus the target function $y$ becomes:

$$y = r_k + \gamma \min_{j=1,2} Q_{\theta_j}(s_{k+1}, a_{k+1}). \tag{32}$$

Results show that this twin delayed double $Q$ network approach has effectively limited overestimation error. As in DDPG, each of the $Q$ values is updated by minimizing the loss function $L(\theta_j)$ with respect to their weights:

$$L(\theta_j) = \mathbb{E}_{s \sim p_\pi, a \sim \pi}[(y - Q_{\theta_j}(s_k, a_k))^2].$$ (33)

The actor network is updated the same way as DDPG (Equation (31)).

3) D4PG [6] is based on DDPG and they utilizes "$n$-step" returns with "$n$-step" horizon when estimating the TD error and replacing the Bellman operator with "$n$-step" variant:

$$\begin{aligned}(\mathcal{T}_\pi^n Q)(x_k, a_k) = r_k + \mathbb{E}[\sum_{t=k}^{k+n-1} \gamma^t r_t \\ + \gamma^n Q(x_{k+n}, \pi(x_{k+n})) \mid x_k, a_k]\end{aligned}$$ (34)

where $(\mathcal{T}_\pi^n Q)(\mathbf{x}_k, \mathbf{a}_k)$ is the distributional $Q$-value function they proposed in [6] and the expectation is with respect to the "$n$-step" transition dynamics.

4) PPO [4] is an on-policy policy gradient method that learns the state value function $V(s_k)$. Based on a fixed length ($T$) horizon, in stead of directly use $V(s_k)$ in Bellman equation, it uses a truncated version of a generalized advantage estimation (GAE) of $A_k$ [21]:

$$\begin{aligned}\hat{A}_k &= \delta_k + (\gamma\lambda)\delta_{k+1} + \cdots + \cdots + (\gamma\lambda)^{T-k+1}\delta_{T-1}, \\ \delta_k &= r_k + \gamma V(s_{k+1}) - V(s_k),\end{aligned}$$ (35)

where $\lambda$ is the GAE parameter [4] to compromise between estimation variance and bias.

5) The mean reward method [35] utilizes "$n$-step" horizon information based on the following mean reward,

$$r_{avg} = \frac{1}{n}(\sum_{t=k}^{k+n-1} r_t),$$ (36)

which is used to update the Bellman equation as in a single step method,

$$Q^\pi(s_k, a_k) = r_{avg} + \gamma Q^\pi(s_{k+1}, a_{k+1}).$$ (37)

## Appendix E   Additional Detailed Results

Here we provide additional detailed results to supplement those that we have reported in the paper. This is a complete set of results for all experiments.

### E.1   Systematic Evaluation of LNSS without Reward Noise.

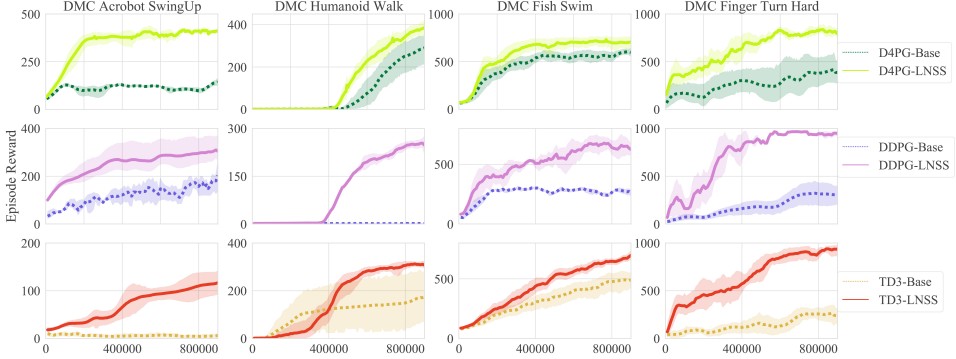

Figure 7: Systematic evaluation of LNSS piggybacked on three DRL algorithms (DDPG, D4PG, TD3) in DMC environments without reward noise. The shaded regions represent the 95 % confidence range of the evaluations over 10 seeds. The x-axis is the number of steps.

## E.2 Results with noise level 1%

Here we include a detailed performance evaluation at noise level of 1% in Figure 8 and Figure 9.

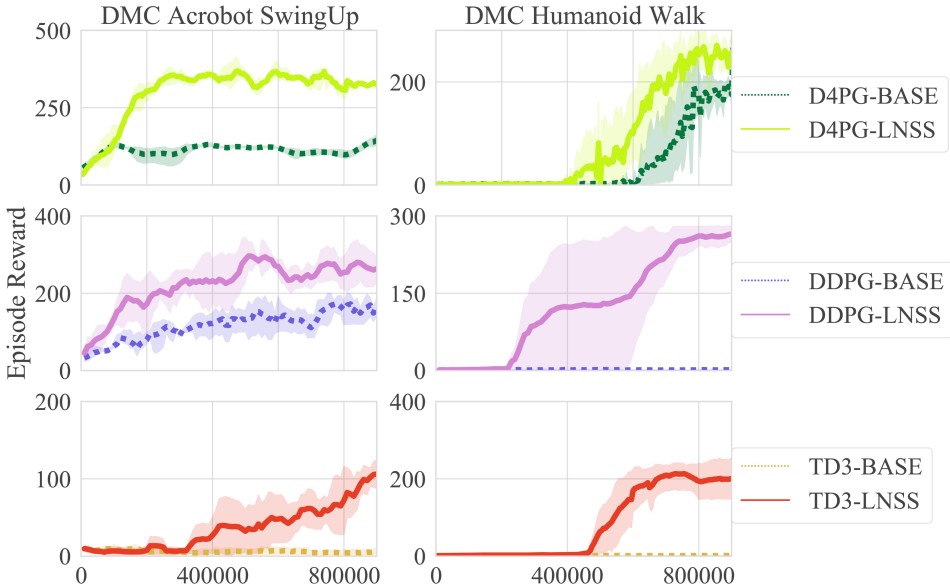

Figure 8: Systematic evaluation of LNSS piggybacked on three DRL algorithms (DDPG, D4PG, TD3) in DMC environments with 1% reward noise. The shaded regions represent the 95 % confidence range of the evaluations over 10 seeds. The x-axis is the number of steps.

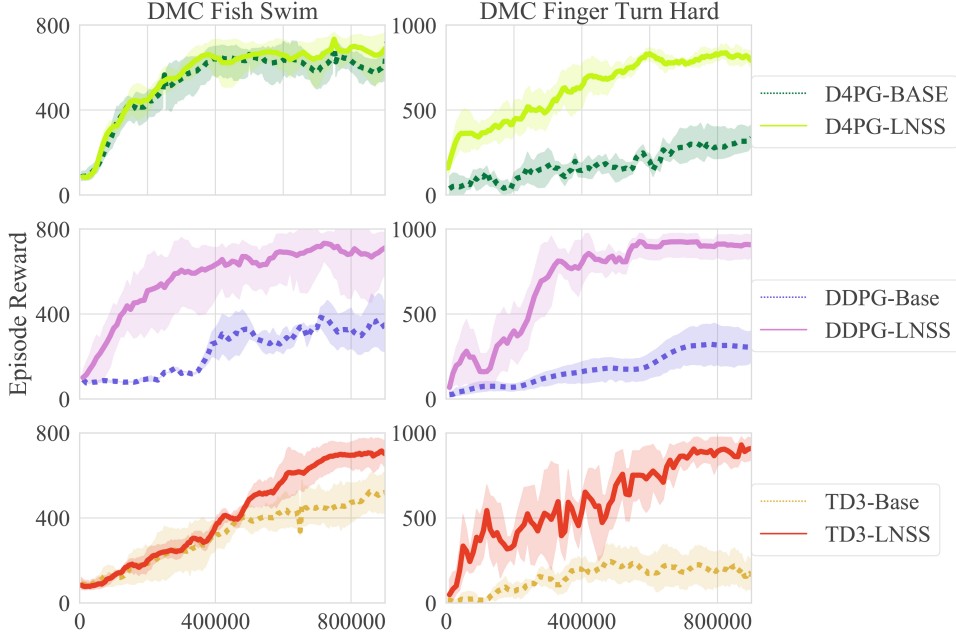

Figure 9: Systematic evaluation of LNSS piggybacked on three DRL algorithms (DDPG, D4PG, TD3) in DMC environments with 1% reward noise. The shaded regions represent the 95 % confidence range of the evaluations over 10 seeds. The x-axis is the number of steps.

## E.3 Results with noise level 10%

Here we include a detailed performance evaluation at noise level of 10% in Figure 10 and Figure 11

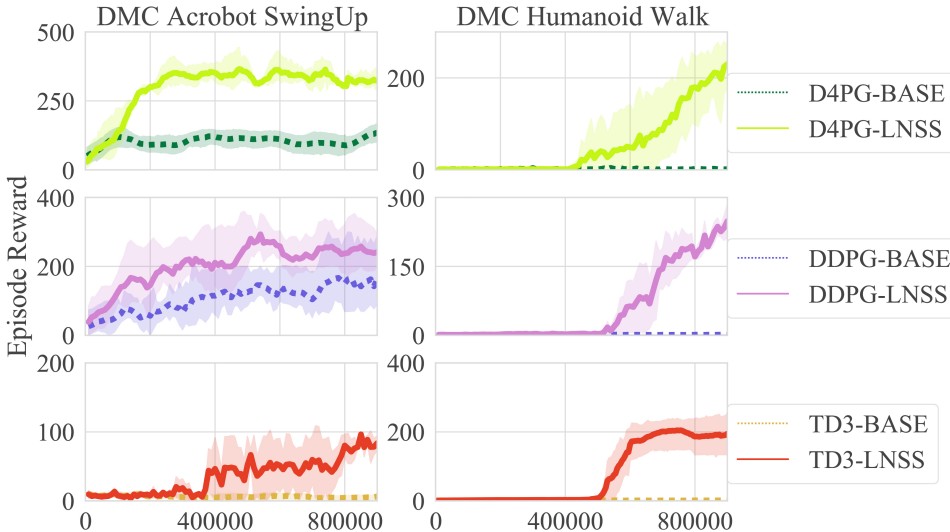

Figure 10: Systematic evaluation of LNSS piggybacked on three DRL algorithms (DDPG, D4PG, TD3) in DMC environments with 10% reward noise. The shaded regions represent the 95 % confidence range of the evaluations over 10 seeds. The x-axis is the number of steps.

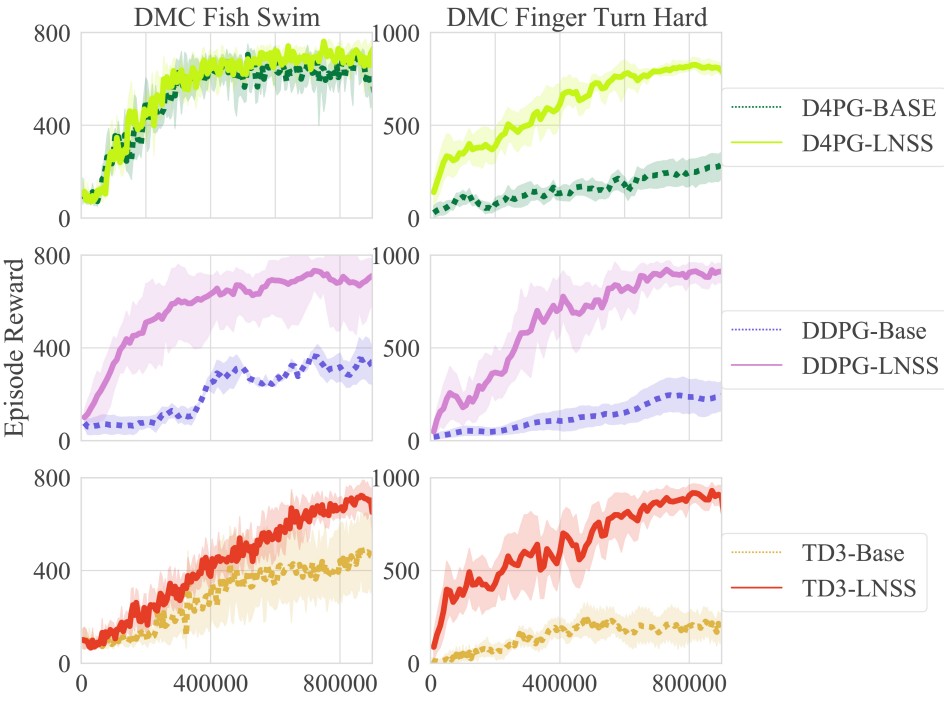

Figure 11: Systematic evaluation of LNSS piggybacked on three DRL algorithms (DDPG, D4PG, TD3) in DMC environments with 10% reward noise. The shaded regions represent the 95 % confidence range of the evaluations over 10 seeds. The x-axis is the number of steps.

## E.4 Variance Study at different levels of reward noise

**Note in the following results (Figure 12 to Figure 20), "Best" denotes the best performing algorithm under all evaluation conditions for a given task. Please refer to Table 2 for correspondence.**

**Without Reward noise results:**

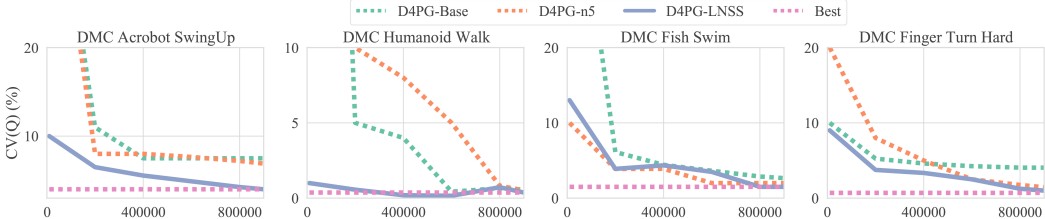

Figure 12: $cv(Q)$ (coefficient of variation of $Q$ value, the lower the better) of D4PG in DMC environments. Results are averaged over 10 seeds. The x-axis is the number of steps.

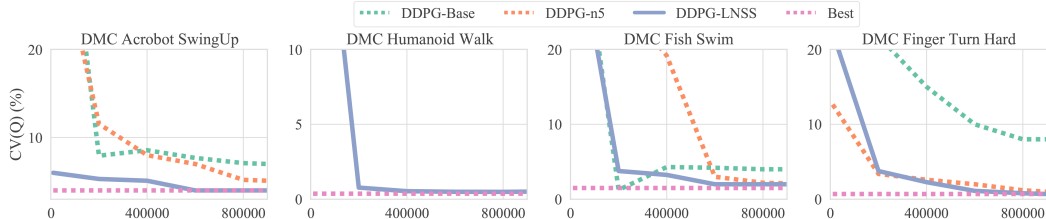

Figure 13: $cv(Q)$ (coefficient of variation of $Q$ value, the lower the better) of DDPG in DMC environments. Results are averaged over 10 seeds. The x-axis is the number of steps.

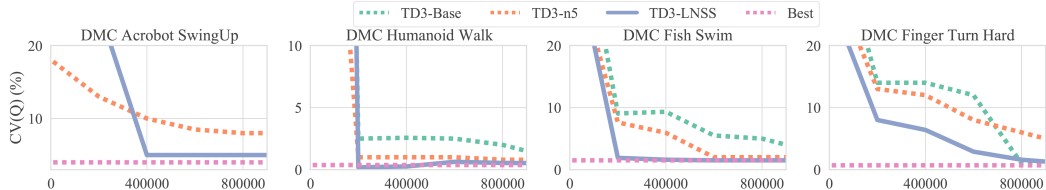

Figure 14: $cv(Q)$ (coefficient of variation of $Q$ value, the lower the better) of TD3 in DMC environments. Results are averaged over 10 seeds. The x-axis is the number of steps.

**1% Reward noise results:**

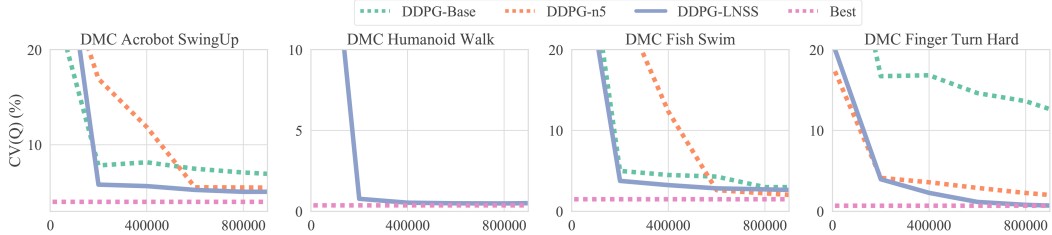

Figure 15: $cv(Q)$ (coefficient of variaiton of $Q$ value, the lower the better) of D4PG in DMC environments. Results are averaged over 10 seeds. The x-axis is the number of steps.

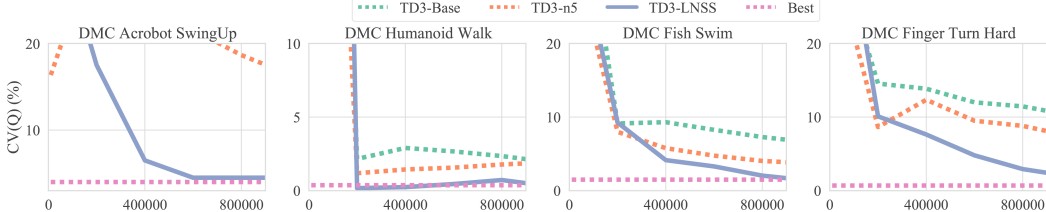

Figure 16: $cv(Q)$ (coefficient of variation of $Q$ value, the lower the better) of DDPG DMC environments. Results are averaged over 10 seeds. The x-axis is the number of steps.

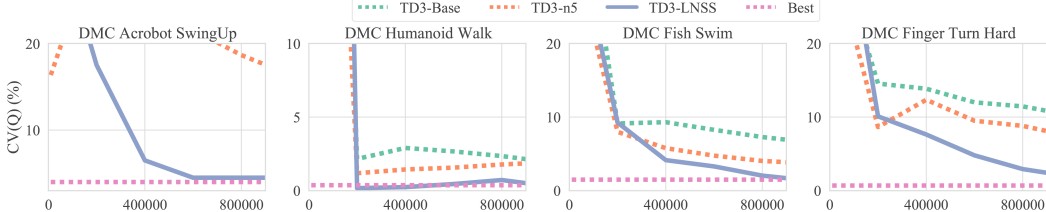

Figure 17: $cv(Q)$ (coefficient of variation of $Q$ value, the lower the better) of TD3 in DMC environments. Results are averaged over 10 seeds. The x-axis is the number of steps.

**10% Reward noise results:**

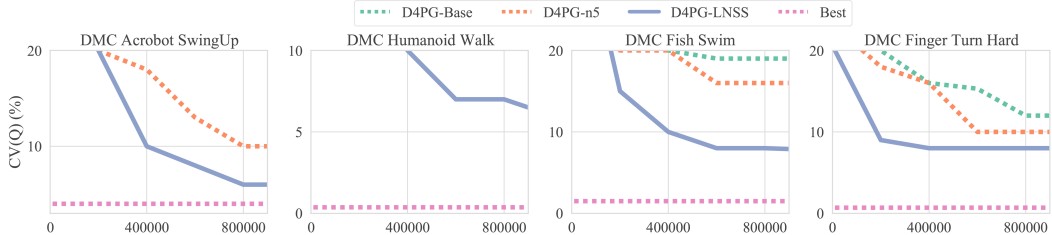

Figure 18: $cv(Q)$ (coefficient of variation of $Q$ value, the lower the better) of D4PG in DMC environments.. Results are averaged over 10 seeds. The x-axis is the number of steps.

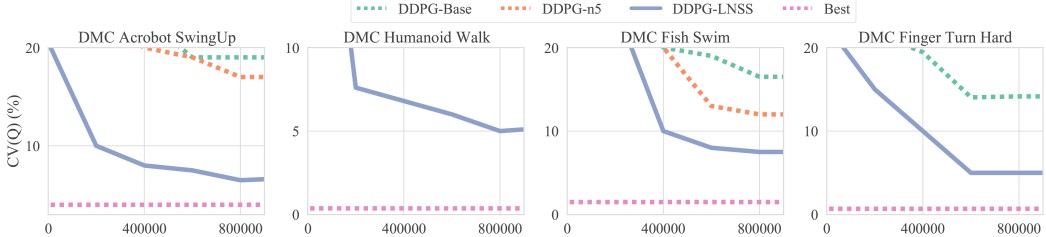

Figure 19: $cv(Q)$ (coefficient of variation of $Q$ value, the lower the better) of DDPG in DMC environments.. Results are averaged over 10 seeds. The x-axis is the number of steps.

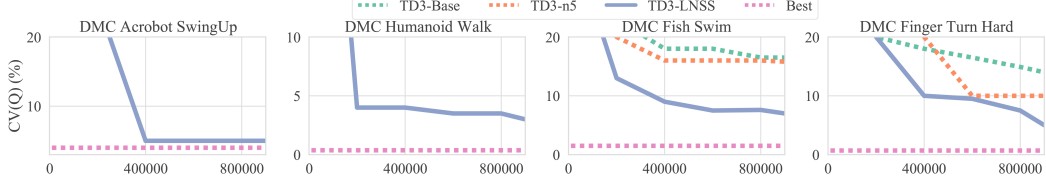

Figure 20: $cv(Q)$ (coefficient of variation of $Q$ value, the lower the better) of TD3 in DMC environments.. Results are averaged over 10 seeds. The x-axis is the number of steps.

### E.5 Gym Evaluation

In Figure 21, we evaluate LNSS on top of TD3 using several challenging continuous control tasks in Gym under a noise-free setting. Here we can see LNSS outperformed its Base method in Walker2D and Hopper but in Humanoid, it only matches its base method. One of the reasons is early termination. Different from DMC tasks, Gym allows an environment to terminate before reaching the maximum time steps. As such, in early training stage, only 10 or 20 steps may be used to compute LNSS which diminishes the power of large N (such as 50 or 100).

Additionally, LNSS is much stable than n5 method in Gym and TD3-n5 method lost its performance improvement in Gym environments. One potential reason is that different from DMC tasks, Gym does not have a uniform reward range with a maximum reward much greater than 1. Therefore, as we mentioned before, in n5 method, the large scale of "$n$-step" returns can cause saturation and inefficiency in learning [10]. However, LNSS has the same reward scales as Base method even with longer $N$ which prevent the potential large reward scale problem.

### E.6 LNSS vs. mean reward method

In Figure 22, we compare performance of TD3 with mean reward method using long reward sequences ($n = 100$ in Equation (36)) in all 6 tasks in Gym and DMC. The figure shows, with LNSS reward,

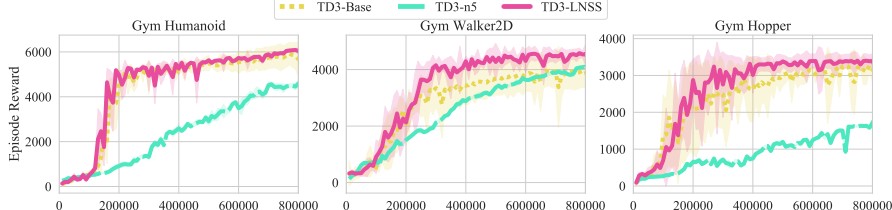

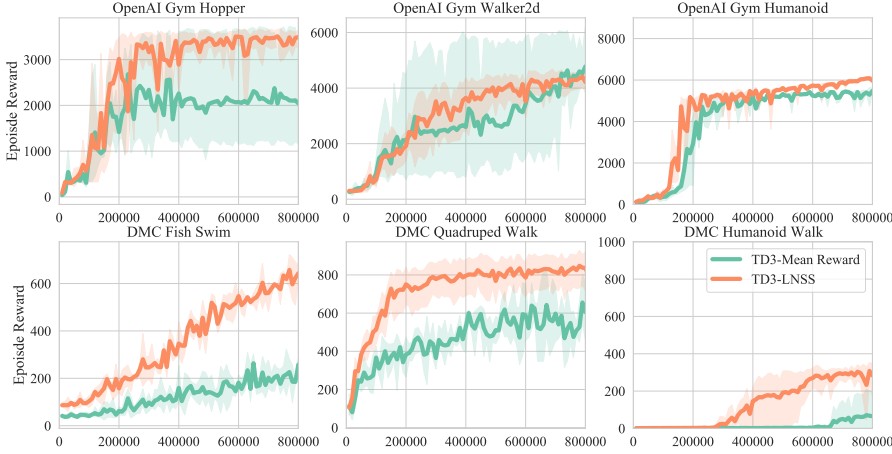

Figure 21: Systematic evaluation in GYM environments of LNSS on top of TD3 using several challenging continuous control tasks in Gym under noise-free setting. The shaded regions represent half a standard deviation of the average evaluation over 10 trials. The x-axis of the plots is the number of steps.Due to early termination in GYM, LNSS was not taking full effect in Humanoid Task. For ramifications refer to the Limitation section in the paper.

Figure 22: Overall performance curves of LNSS and mean reward method over all tasks. The shaded region represents half a standard deviation of the average evaluation over 10 trials.

TD3-LNSS outperforms the mean reward method in all tasks in terms all measures (episode reward, learning variance, and learning speed).

One potential reason for mean reward method cannot address return with a long horizon is that as in Equation (36)), $r_{avg}$ treats every stage reward with the same significance (without discount) which will hurt learning. For example, when considering a reward trajectory with $n >> 1$, the first step should have more significance than the reward at $n$ step.

## E.7  Hyper-parameter with different $N$ Comparison

In Figure 23, we show the effect of different choices of $N$ ($N = 5, 10, 20, 50, 100$) in our proposed LNSS for D4PG. Comparisons are performed using the same set of hyperparameters except that $N$ is different in all experiments. The significance of large $N$ value varies task to task. In the DMC environment, $N = 100$ outperforms others in reward and learning speed and matches others in learning variance.

In Figure 24, we show the effect of different choices of $N$ ($N = 5, 10, 20, 50, 100$) in our proposed LNSS for DDPG. Comparisons are performed using the same set of hyperparameters except that $N$ is different in all experiments. The significance of large $N$ value varies task to task. In the DMC environment, $N = 100$ outperforms others in learning speed and variance and matches others in episode reward.

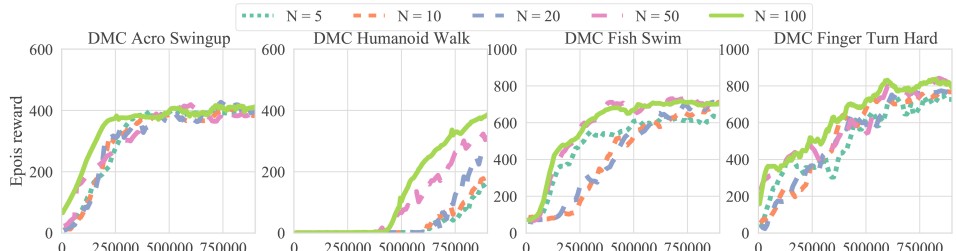

Figure 23: Episode rewards for the 4 tasks in DMC by D4PG-LNSS with different $N$ ($N = 5, 10, 20, 50, 100$) in Equation (6). The results are averaged from 10 seeds. The x-axis is the number of steps.

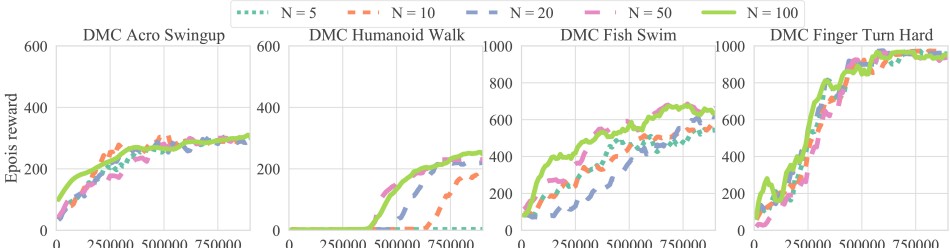

Figure 24: Episode rewards for the 4 tasks in DMC by DDPG-LNSS with different $N$ ($N = 5, 10, 20, 50, 100$) in Equation (6). The results are averaged from 10 seeds. The x-axis is the number of steps.