# OpenReview forum: "A Long $N$-step Surrogate Stage Reward for Deep Reinforcement Learning"
_NeurIPS.cc/2023/Conference — NeurIPS 2023 poster_

### Official Review · Reviewer_r58G · 2023-06-21

**Soundness:** 2 fair
**Presentation:** 3 good
**Contribution:** 2 fair
**Rating:** 5
**Confidence:** 3

**Summary:**

The article considers a new way to formulate the n-step rewards in deep reinforcement learning. Specifically, while the normal n-step reward discount the Q-value by \gamma^n, the authors propose to only discount the Q-value by \gamma while using the normal n-step reward (times a constant) for r. For a simplified model, the authors mathematically prove that this decreases an upper bound of the variance of Q-values. Empirically, the authors add the proposed method to three standard RL algorithms (DDPG, TD3, D4PG) and evaluate them on DMC and GYM. For four tasks in DMC, the proposed method improves the performance of the baseline algorithms. For GYM, the proposed methods improve TD3 on a task.

EDIT: I have read the rebuttal and will keep my score but decrease my certainty.

**Strengths:**

The paper proposes a natural and simple method. It should be simple to implement and could conceivably improve performance.

The authors show strong empirical performance for the proposed method.

**Weaknesses:**

The authors show improvements on a few tasks for a few algorithms. It is unclear how the environments are selected, and there is some risk of cherry-picking. Furthermore, it is not clear if the baseline algorithms and their implementations are SOTA. So it’s not really clear if the proposed methods would improve SOTA RL, they might just improve some mediocre baselines.

The assumptions of theorem 1 are not clear. Furthermore, I think the interpretation of the results is misleading. The authors show that an upper bound on the variance of the Q-values decreases with the proposed method. However, a difference in an upper bound does NOT imply that the actual variance decreases. That is only true if the upper bound is reasonably tight.

**Questions:**

Are the experiments all state-based? If so, could you add experiments when learning from pixels?

How were the environments selected?

Will the upper bound in Theorem 1 be tight?

Can you clarify the assumptions in Theorem 1?

Are the implementations SOTA? If not, can you try adding your method to a SOTA implementation?

---

> ### Author Rebuttal · Authors · 2023-08-09
>
> We greatly appreciate the reviewer's comments and suggestions. We'll properly address the questions in our revision if the paper is accepted.
>
> >Q1. The author selected a few tasks for a few algorithms
>
> Thank you for this important comment. We believe that the number of tasks and the baseline algorithms we selected in our paper is in line with the most recent, SOTA results published in top-tier conferences such as NeuIPS 2022 and ICML 2022. For example, for evaluations in [1-5], the authors used from 2 to 5 benchmark environments. They used three baseline algorithms. Together with their 6 variant algorithms. they evaluated between 3 to 9 algorithms.
>
> In comparison, our evaluations involved 7 different DMC benchmark environments for all base algorithms in our main paper, and additional 3 different GYM benchmark environments for TD3. For algorithm selection, as in those papers [1-5], we used 3 baseline algorithms. For each algorithm, we have 2 different variants (n-step and LNSS). This makes our total compared algorithms to be 9.
>
> In sum, we believe our selection of benchmark environments and baseline algorithms was on par with the most recent comparable papers in top venues.
>
> >Q2. How were the environments selected?
>
> We kept a general idea in mind that is to diversify our benchmark environments. We not only select dense reward environments, but also sparse reward environments. Additionally, We considered reward noise in each dense reward environment to make the  environments more realistic (more often than not, previous works do not consider reward noise).
>
> Among all DMC benchmark environments, the four types of environments are classical control dense, classical control sparse, robot locomotion dense, and robot locomotion sparse. For classical control types, many recent papers choose carpole swingup [1,6,7,8] which is much easier to learn than acrobot-swingup as discussed in [9,12]. We thus chose acrobot-swingup. For robot locomotion, the complexity of the environment increases as the dimension of the state and action space increases. Many recent papers use walker-walk [1,6,8,10,13] with  state space dimension of 18 and action space 6, and cheetah-run [4,9,11,13,16] with state space 18 and action space 6. These two environments are much easier to learn than our selection of humanoid-walk [1,8,9] with state space dimension of 54 and action space 21. Additionally, fish-swim [10,12] and finger-turn hard [10,11,12] are also commonly selected locomotion benchmark environments. For classic control sparse type, we use carpole swingup sparse [8,13], a commonly selected sparse benchmark. For locomotion sparse type, as other papers [14,15], we used cheetah and walker to sparsify the dense reward, a common practice in literature [14,15]. We therefore have a locomotion sparse benchmark. For GYM environments, Hopper, walker2d and humanoid are the common selection for evaluations in the literature [12,16,17]. We therefore did the same.
>
> >Q3. Are baseline and its implementation SOTA?
>
> Based on comprehensive benchmark results on Deepmind Control Suite [9,18,19], our choice of off-policy baselines are SOTA. Specifically, it is well accepted that D4PG provides SOTA results in continuous control problems. This can also be seen from the highest rewards achieved by D4PG with scores over 900 over 1000. As such D4PG has been used in most of the recent and impactful papers [1,20,23,24]. Although DDPG and TD3 do not perform as well as D4PG, they still are considered SOTA in RL as they are widely used as baseline algorithms in recent impactful papers [2,3,21,22,23].
>
> Additionally, notice from Table 1 that, LNSS has enabled DDPG and TD3 to achieve comparable performance as SOTA D4PG. This also demonstrates the significance and impact of LNSS and why it is a novel and practical idea.
>
> >Q4. Clarify the assumptions in Theorem 1? and will the upper bound in Theorem 1 be tight?
>
> We assume state-action value function $Q$ and $\mathbb{Q}$ are initialized to 0 ($Q_0 = 0$ and $\mathbb{Q}_0 = 0$) which is a common assumption and practically used in reinforcement learning [25,26]. Accordingly,  the variance of initial $Q$ values are 0, i.e., $Var[Q_0]=0$ and $Var[\mathbb{Q}_0]=0$
>
> We assume the reward samples follow IID process which is a common and widely used assumption in reinforcement learning [27,28]
>
> We Assume the variance of $\{r_k\}$ is upper bounded by a finite positive number $\mathbb{B}$, i.e., $Var[r_k] \leq \mathbb{B}$. This is widely used and practical. Consider a reward bounded by $[minR,maxR]$, the maximum variance can be calculated using Popoviciu's inequality: $Var[r] \leq \frac{1}{4}(maxR-minR)^2$. Hence our  $\mathbb{B} = \frac{1}{4}(maxR-minR)^2 $. For DMC environments, the rewards are within [0,1]. Other environments such as Atari domain clips reward to be within [0,1]. Therefore in many cases,  $\mathbb{B} = \frac{1}{4}$.
>
> LNSS theoretically reduces the upper bound of variance according to Theorem 1. Furthermore, this bound exponentially decreases as N increases. For example, for N=100, Refer to Remark 2 and Figure 1, we can see that $\psi=0.011$. Therefore, in this case, LNSS has reduced the upper bound of variance by about 100 times from single-step methods. Our extensive evaluation results also show significantly reduced variance due to LNSS. (Table 1 and Fig 4, 6 and all figures in Appendix)
>
> >Q5. Are the experiments all state-based? If so, could you add experiments when learning from pixels?
>
> Yes, this paper is about solving state-based problems. Probably for similar reasons, most works [1-8,10-17,19,21-29] also mainly focus state-based problems. Actually, the literature covers either state-based or directly pixel-based problems, not both. In principle, we expected LNSS to be effective for pixel-based problems, in which case, we will need to use deep NN/ CNN in place of our fully connected networks. We are more than happy to quantitively evaluate LNSS on pixel-based problems in our future work.

---

> > ### Author Response · Authors · 2023-08-10
> > **Reference For Above Rebuttal**
> >
> >
> > [1] Huang, S., Abdolmaleki, A., Vezzani, G., Brakel, P., Mankowitz, D. J., Neunert, M., \& Hadsell, R. (2022, January). A constrained multi-objective reinforcement learning framework. In Conference on Robot Learning (pp. 883-893). PMLR.
> >
> > [2] Cetin, E., \& Celiktutan, O. (2021, July). Learning routines for effective off-policy reinforcement learning. In International Conference on Machine Learning (pp. 1384-1394). PMLR.
> >
> > [3] Wilcox, A., Balakrishna, A., Dedieu, J., Benslimane, W., Brown, D., \& Goldberg, K. (2022). Monte carlo augmented actor-critic for sparse reward deep reinforcement learning from suboptimal demonstrations. Advances in Neural Information Processing Systems, 35, 2254-2267.
> >
> > [4] Devidze, R., Kamalaruban, P., \& Singla, A. (2022). Exploration-Guided Reward Shaping for Reinforcement Learning under Sparse Rewards. Advances in Neural Information Processing Systems, 35, 5829-5842.
> >
> > [5] MacGlashan, J., Archer, E., Devlic, A., Seno, T., Sherstan, C., Wurman, P., \& Stone, P. (2022). Value Function Decomposition for Iterative Design of Reinforcement Learning Agents. Advances in Neural Information Processing Systems, 35, 12001-12013.
> >
> > [6] Yue, Y., Kang, B., Xu, Z., Huang, G., \& Yan, S. (2023, June). Value-Consistent Representation Learning for Data-Efficient Reinforcement Learning. In Proceedings of the AAAI Conference on Artificial Intelligence (Vol. 37, No. 9, pp. 11069-11077).
> >
> > [7] Liu, X., Yoneda, T., Wang, C., Walter, M., \& Chen, Y. (2023, July). Active Policy Improvement from Multiple Black-box Oracles. In International Conference on Machine Learning (pp. 22320-22337). PMLR.
> >
> > [8] Seyde, T., Gilitschenski, I., Schwarting, W., Stellato, B., Riedmiller, M., Wulfmeier, M., \& Rus, D. (2021). Is bang-bang control all you need? solving continuous control with bernoulli policies. Advances in Neural Information Processing Systems, 34, 27209-27221.
> >
> > [9] Tassa, Y., Doron, Y., Muldal, A., Erez, T., Li, Y., Casas, D. D. L., ... \& Riedmiller, M. (2018). Deepmind control suite. arXiv preprint arXiv:1801.00690.
> >
> > [10] Cetin, E., \& Celiktutan, O. (2021, July). Learning routines for effective off-policy reinforcement learning. In International Conference on Machine Learning (pp. 1384-1394). PMLR.
> >
> > [11] Chang, J., Wang, K., Kallus, N., \& Sun, W. (2022, June). Learning bellman complete representations for offline policy evaluation. In International Conference on Machine Learning (pp. 2938-2971). PMLR.
> >
> > [12] Li, Q., Kumar, A., Kostrikov, I., \& Levine, S. (2023). Efficient Deep Reinforcement Learning Requires Regulating Overfitting. arXiv preprint arXiv:2304.10466.
> >
> > [13] Stooke, A., Lee, K., Abbeel, P., \& Laskin, M. (2021, July). Decoupling representation learning from reinforcement learning. In International Conference on Machine Learning (pp. 9870-9879). PMLR.
> >
> > [14] Zhang, T., Ren, T., Yang, M., Gonzalez, J., Schuurmans, D., \& Dai, B. (2022, June). Making linear mdps practical via contrastive representation learning. In International Conference on Machine Learning (pp. 26447-26466). PMLR.
> >
> > [15] Rengarajan, D., Chaudhary, S., Kim, J., Kalathil, D., \& Shakkottai, S. (2022). Enhanced Meta Reinforcement Learning via Demonstrations in Sparse Reward Environments. Advances in Neural Information Processing Systems, 35, 2737-2749.
> >
> > [16] Yin, Z. H., Ye, W., Chen, Q., \& Gao, Y. (2022). Planning for Sample Efficient Imitation Learning. Advances in Neural Information Processing Systems, 35, 2577-2589.
> >
> > [17] Yang, Q., Wang, S., Lin, M. G., Song, S., \& Huang, G. (2023). Boosting Offline Reinforcement Learning with Action Preference Query. arXiv preprint arXiv:2306.03362.

---

> > > ### Author Response · Authors · 2023-08-10
> > > **cont**
> > >
> > > [18] Pardo, F. (2020). Tonic: A deep reinforcement learning library for fast prototyping and benchmarking. arXiv preprint arXiv:2011.07537.
> > >
> > > [19] Barth-Maron, G., Hoffman, M. W., Budden, D., Dabney, W., Horgan, D., Tb, D.,  \& Lillicrap, T. (2018). Distributed distributional deterministic policy gradients. arXiv preprint arXiv:1804.08617.
> > >
> > > [20] Gulcehre, C., Wang, Z., Novikov, A., Paine, T., Gómez, S., Zolna, K., \& de Freitas, N. (2020). Rl unplugged: A suite of benchmarks for offline reinforcement learning. Advances in Neural Information Processing Systems, 33, 7248-7259.
> > >
> > > [21] Sun, H., Han, L., Yang, R., Ma, X., Guo, J., \& Zhou, B. (2022). Exploit Reward Shifting in Value-Based Deep-RL: Optimistic Curiosity-Based Exploration and Conservative Exploitation via Linear Reward Shaping. Advances in Neural Information Processing Systems, 35, 37719-37734.
> > >
> > > [22] Chakraborty, S., Bedi, A. S., Tokekar, P., Koppel, A., Sadler, B., Huang, F., \& Manocha, D. (2023, June). Posterior coreset construction with kernelized stein discrepancy for model-based reinforcement learning. In Proceedings of the AAAI Conference on Artificial Intelligence (Vol. 37, No. 6, pp. 6980-6988).
> > >
> > > [23] Agarwal, R., Schwarzer, M., Castro, P. S., Courville, A. C., \& Bellemare, M. (2022). Reincarnating reinforcement learning: Reusing prior computation to accelerate progress. Advances in Neural Information Processing Systems, 35, 28955-28971.
> > >
> > > [24] Chen, X., Mu, Y. M., Luo, P., Li, S., \& Chen, J. (2022, June). Flow-based recurrent belief state learning for pomdps. In International Conference on Machine Learning (pp. 3444-3468). PMLR.
> > >
> > >
> > > [25] Fujimoto, S., Hoof, H., \& Meger, D. (2018, July). Addressing function approximation error in actor-critic methods. In International conference on machine learning (pp. 1587-1596). PMLR.
> > >
> > > [26] Sutton, R. S., \& Barto, A. G. (2018). Reinforcement learning: An introduction. MIT press.
> > >
> > > [27] Marom, O., \& Rosman, B. (2018, April). Belief reward shaping in reinforcement learning. In Proceedings of the AAAI conference on artificial intelligence (Vol. 32, No. 1).
> > >
> > > [28] Lu, X., \& Van Roy, B. (2017). Ensemble sampling. Advances in neural information processing systems, 30.
> > >
> > > [29] Henderson, P., Islam, R., Bachman, P., Pineau, J., Precup, D., \& Meger, D. (2018, April). Deep reinforcement learning matters. In Proceedings of the AAAI conference on artificial intelligence (Vol. 32, No. 1).

---

> > > > ### Comment · Reviewer_r58G · 2023-08-12
> > > > **reply**
> > > >
> > > > Thanks for your reply!
> > > >
> > > > Regarding baselines, I am interested in knowing if the "implementation" of the algorithms you use are SOTA. Not only the algorithms themselves. In RL, implementations vary a lot even for the same algorithms, and subtle implementation tricks can have large effects. Were the implementations themselves taken from some SOTA library? Or did you implement these yourselves?
> > > >
> > > > Furthermore, regarding the upper bound -- do you have any mathematical proof regarding how tight the bound is?

---

> > > > > ### Author Response · Authors · 2023-08-20
> > > > > **Are Implementations of Algorithms SOTA?**
> > > > >
> > > > > >Regarding baselines, I am interested in knowing if the "implementation" of the algorithms you use are SOTA. Not only the algorithms themselves. In RL, implementations vary a lot even for the same algorithms, and subtle implementation tricks can have large effects. Were the implementations themselves taken from some SOTA library? Or did you implement these yourselves?
> > > > >
> > > > > Thank you for the clarification on "SOTA implementation" rather than the algorithms themselves.
> > > > >
> > > > > 1. All implementation details (including hyperparameters, seed selection, network structure, etc.) are all directly from github repositories. Please refer to our Appendix B.
> > > > >
> > > > > 2. For the simulation environments, the DMC package was installed under the instructions that Deepmind provided in their github repository.
> > > > >
> > > > > 3. For DDPG and D4PG, we chose all hyperparameters the same as in the DMC paper [R2], which corresponds to the best-reported results [R2]. Due to our physical limitation in computing resources, we use 8 or 20 parallel cores in evaluation while D4PG [R3] and DMC paper [R2] reported  use of 32 or 64 cores. Since TD3 was not included in [R2], we chose to use exact same hyperparameters as we used in DDPG and D4PG. All hyperparameters of DDPG, TD3 and D4PG are summarized in Appendix B tables 2, 3, 4, respectively.
> > > > >
> > > > > 4. The TD3 and DDPG codes are directly from the TD3 paper authors [R4] (They provided a fine tuned DDPG) as described in our Appendix B where we provide the link to the GitHub repositories of sfujim. For D4PG, since the author does not provide a code, we implement it based on TD3 as described in Appendix B.
> > > > >
> > > > > 5. For obtaining evaluation results, we make sure all the comparisons are fair, apple-to-apple, without altering any hyper-parameters. The hyperparameters are the same in baseline algorithms, LNSS and n-step.
> > > > >
> > > > > 6. As promised in our paper, we will release all codes if the paper is accepted.
> > > > >
> > > > > [R1] Henderson, P., Islam, R., Bachman, P., Pineau, J., Precup, D., \& Meger, D. (2018, April). Deep reinforcement learning matters. In Proceedings of the AAAI conference on artificial intelligence (Vol. 32, No. 1).
> > > > >
> > > > > [R2] Tassa, Y., Doron, Y., Muldal, A., Erez, T., Li, Y., Casas, D. D. L., ... \& Riedmiller, M. (2018). Deepmind control suite. arXiv preprint arXiv:1801.00690.
> > > > >
> > > > > [R3] Barth-Maron, G., Hoffman, M. W., Budden, D., Dabney, W., Horgan, D., Tb, D.,  \& Lillicrap, T. (2018). Distributed distributional deterministic policy gradients. arXiv preprint arXiv:1804.08617.
> > > > >
> > > > > [R4] Fujimoto, S., Hoof, H., \& Meger, D. (2018, July). Addressing function approximation error in actor-critic methods. In International conference on machine learning (pp. 1587-1596). PMLR.

---

> > > > > > ### Author Response · Authors · 2023-08-20
> > > > > > **Is the Bound on Variance Tight?**
> > > > > >
> > > > > > >Furthermore, regarding the upper bound -- do you have any mathematical proof regarding how tight the bound is?
> > > > > >
> > > > > > Consider two $Q$ values using the original reward and LNSS method. We place the two results, equations (15) and (16), from the paper on bounds of variances here:
> > > > > >
> > > > > > $Var[Q_{i+1}(s_{k},a_{k})] \leq \sum_{t=1}^{i+1}(\gamma^{t-1})^2\mathbb{B}, $  --- Eqn (R7)
> > > > > >
> > > > > > $Var[\mathbb{Q}_{i+1}(s_k,a_k) $
> > > > > >
> > > > > > $\leq  \psi\sum_{t=1}^{i+1}(\gamma^{t-1})^2 \mathbb{B}]$  --- Eqn (R8)
> > > > > >
> > > > > > where $\mathbb{B}$ is maximum variance of stage reward $r_k$, from Eqn (14) in the paper, $\psi = (\frac{\gamma - 1}{\gamma^N - 1})^2(\frac{\gamma^{2N} - 1}{\gamma^2 - 1})$, and $i$ is the iteration number.
> > > > > >
> > > > > >
> > > > > > **Lemma R1.** Assume stage reward $r_k$ is within $[0, R_{max}]$, $R_{max} > 0$. Then the variance of $r_k$ is bounded by $Var[r_k] \leq \mathbb{B} = \frac{1}{4}(R_{max})^2$.
> > > > > >
> > > > > > Proof. By applying Popoviciu's inequality, $Var[r_k] \leq \frac{1}{4}(R_{max} - 0)^2$.
> > > > > >
> > > > > > **Theorem R2.** Consider the variances of two $Q$ value sequences, denoted as {${Q_i}$} and
> > > > > > {$\mathbb{Q}_i$}
> > > > > >
> > > > > > in Equation (R7) and Equation (R8), which are obtained respectively from an original reward and an LNSS method. Let the IID reward $\{r_k\}$ and $\{r_k'\}$ be drawn
> > > > > > from the memory buffer and bounded by $[0, R_{max}]$, $R_{max} > 0$. Then the  upper bounds of the variances of the two $Q$ value sequences, as iteration $i \to \infty$,
> > > > > > ${Var[Q_{\infty}]}$ and $Var[\mathbb{Q}_{\infty}]$, are respectively described below,
> > > > > >
> > > > > >
> > > > > > $Var[Q_{\infty}(s_{k},a_{k})] \leq \frac{1}{1 - \gamma^2} \frac{1}{4}(R_{max})^2,$  --- Eqn (R9)
> > > > > >
> > > > > >
> > > > > > $Var[\mathbb{Q}_{\infty}(s_k,a_k)] $
> > > > > >
> > > > > > $\leq \psi \frac{1}{1 - \gamma^2} \frac{1}{4}(R_{max})^2.$  --- Eqn (R10)
> > > > > >
> > > > > >
> > > > > >
> > > > > > Proof. From Lemma R1, the upper bound of the original reward method (Eqn R7) can be rewritten as
> > > > > >
> > > > > > $Var[Q_{i+1}(s_{k},a_{k})] \leq \sum_{t=1}^{i+1}(\gamma^{t-1})^2 \frac{1}{4}(R_{max})^2$  --- Eqn (R11)
> > > > > >
> > > > > > $Var[Q_{i+1}(s_{k},a_{k})] \leq \frac{1 - \gamma^{2i + 2}}{1 - \gamma^2} \frac{1}{4}(R_{max})^2$  --- Eqn (R12)
> > > > > >
> > > > > > with $i \to \infty$, $Var[Q_{\infty}(s_{k},a_{k})] \leq \frac{1}{1 - \gamma^2} \frac{1}{4}(R_{max})^2$.
> > > > > >
> > > > > > Then from Lamma R1, the upper bound of LNSS method (Eqn R8) can be rewritten as
> > > > > >
> > > > > > $Var[\mathbb{Q}_{i+1}(s_k,a_k)] $
> > > > > >
> > > > > > $\leq \psi \frac{1 - \gamma^{2i + 2}}{1 - \gamma^2} \frac{1}{4}(R_{max})^2$. --- Eqn (R13)
> > > > > >
> > > > > > With $i \to \infty$,
> > > > > >
> > > > > > $Var[\mathbb{Q}_{\infty}(s_k,a_k)]$
> > > > > >
> > > > > > $\leq \psi \frac{1}{1 - \gamma^2} \frac{1}{4}(R_{max})^2$$.
> > > > > >
> > > > > > Thus Theorem R2 holds.
> > > > > >
> > > > > > **Remark R2.**
> > > > > >
> > > > > > In reinforcement learning, the scale of a reward is usually constrained to a finite or even small range  (e.g., DMC reward is all within [0,1]. Some Atari domain environment also clips reward to be within [0,1]), since a large output scale can result in problems due to saturation and inefficiency in learning [1]. Consider the DMC case with reward bound as $[0,1]$ and in our paper, $\gamma = 0.99$, and our best results using $N = 100$, the upper bound upon convergence can be written as:
> > > > > >
> > > > > >  $Var[Q_{\infty}(s_{k},a_{k})] \leq 12.56,$  --- Eqn (R14)
> > > > > >
> > > > > >  $Var[\mathbb{Q}_{\infty}(s_k,a_k)] \leq 0.138.$  --- Eqn (R15)
> > > > > >
> > > > > > By inspecting Eqn (R14) and Eqn (R15), we see that LNSS has resulted in a much tighter bound on $Q$ value variance than the original method. Also notice that, this bound further decreases exponentially (not increases) with greater $N$ (the reward length). The bound is therefore tight.

---

### Official Review · Reviewer_ektB · 2023-07-05

**Soundness:** 2 fair
**Presentation:** 2 fair
**Contribution:** 2 fair
**Rating:** 4
**Confidence:** 4

**Summary:**

This paper introduces a new stage reward estimator called the long N-step surrogate stage (LNSS) reward for deep reinforcement learning (RL). The LNSS aims to mitigate the high variance problem in deep RL, which has been shown to impede successful convergence of learning, hinder task performance, and limit the application of deep RL in continuous control problems. The paper shows that LNSS, which uses a long reward trajectory of future steps, consistently improves performance in terms of average reward, convergence speed, learning success rate, and variance reduction in Q-values and rewards. The evaluations are based on various environments in DeepMind Control Suite and OpenAI Gym using LNSS in baseline deep RL algorithms such as DDPG, D4PG, and TD3. The paper also demonstrates that LNSS significantly reduces the upper bound on the variances of Q-values compared to single-step methods.

**Strengths:**

The paper proposes a new stage reward estimator, LNSS, which addresses the high-variance problem in RL. This can be useful in improving performance in continuous control tasks.

**Weaknesses:**

1. Compared with the common n-step bootstrap method, it seems that the difference is only the larger n?
2. The paper uses off-policy RL baselines like TD3 and such a large N in LNSS reward, but didn't pay any attention to the off-policy issue.
3. The paper is not well written, making it difficult to identify the originality.

**Questions:**

1. It is obvious that increasing the length of bootstrap can reduce variance, but how do you handle bias (off policy issue)?

**Limitations:**

The paper seems to lack originality.

---

> ### Author Rebuttal · Authors · 2023-08-09
>
> We greatly appreciate the reviewer's critical input, which has helped us to think about LNSS from a general perspective in order to make it more clear and highlight its novelty.
>
> >Q1 compare with common $n$-step method, the difference is only the larger $n$? lack originality
>
> Thank you for the question. We have carefully explained this issue in the General Response, points 1) - 5).
>
> >Q2 How to address the off-policy issue? How do you handle bias (off policy issue)?
>
> Thank you for this important question. Please refer to our General Response, points 7), 5), 6) and 9) where we explained that LNSS does not raise additional estimation bias to the Baseline algorithms
>
> >Q3 The paper is not well written, making it difficult to identify the originality.
>
> We sincerely thank the reviewer for the feedback. Hopefully, with the point to point explanation in our general response, the significance and the novelty of LNSS are now clear so that the entire paper is easy to read now, as other reviewers said in their reviews.

---

### Official Review · Reviewer_TX4g · 2023-07-06

**Soundness:** 3 good
**Presentation:** 4 excellent
**Contribution:** 3 good
**Rating:** 7
**Confidence:** 3

**Summary:**

The paper propose a new reward estimator for off-policy RL to reduce the variance of the Q-value target estimate in bellman backup. Specifically, the method works by computing the discounted return of a trajectory segment and then augment such reward along with the original transition (state, action, next state) in the replay buffer. On a range of simulated environments, the proposed method is able to enhance existing off-policy RL algorithm such as DDPG, D4PG and TD3 on a range of environments in DeepMind Control Suite and OpenAI Gym. The authors also provide variance analysis of the Q-value target estimate and demonstrate an improved variance upper-bound.

**Strengths:**

The paper is well-written and easy to follow. The proposed method is very simple and can be directly added on top of any TD-style RL algorithms. In addition, the paper contains a set of thorough empirical evaluations that demonstrate that the effectiveness and the robustness of the proposed method (LNSS) under a range of reward functions (e.g., noisy, sparse), different hyperparameters as well as insights on the variance reduction effect that LNSS has on the Q value estimation.

**Weaknesses:**

If I understand correctly, the proposed method can lead to a biased value estimate, which conceptually could hurt performance. The bias comes from the fact that the LNSS reward estimate assumes the trajectory is filled with a constant reward. For example, for sparse reward task where the reward is only non-zero, positive in the end of the trajectory, the Q-value would get overestimated. It would be great if the authors could discuss more the implication of the bias and whether there are any pathological examples that could break the method.

**Questions:**

- Have you experimented with using the full trajectory to compute the LNSS reward? I imagine that this could actually help the sparse reward tasks even more (e.g., Cartpole SwingUp Sparse).
- It would be also very interesting to study the interaction of LNSS with more gradient steps per environment step (UTD ratio). Intuitively, having more gradient updates per data collection step could also greatly help with propagating the signal back in time fast (in terms of the number of environment steps).

**Limitations:**

The author has adequately addressed the limitations.

---

> ### Author Rebuttal · Authors · 2023-08-09
>
> We greatly thank the reviewer for reading our paper and for the favorable comments. Your questions have helped us further improve the paper.
>
> > Q1 About estimation bias and learning performance and the LNSS reward estimate assumes the trajectory is filled with a constant reward
>
> Please refer to our General Response, points 7) 5) 6) 9)
>
> About $r_k'$ being a constant reward signal, this may be caused by our inappropriate choice of short hand notation of $G_k'$ in Eqn (5), and thus caused confusion. Actually $G_k'$ in (5) is an imaginary variable in our mind.  It is probably easy to see that it inspired us to reach Eqn (6) by simply moving the sum of discounted factor $\gamma$ away from $r_k'$, and thus we arrive at Eqn (6), for the construction of $r_k'$. We should have used a different notation so that $G_k'$ is as in RL convention that it is the sum of discounted rewards {$r_k'$}.
>
> As for a pathological example that may break LNSS, please refer to our Limitations section where we discussed that if there is a lack of a sufficiently long reward sequence to work with, then LNSS will not be effective.
>
> >Q2 Have you experimented with using the full trajectory to compute the LNSS reward? I imagine that this could actually help the sparse reward tasks even more (e.g., Cartpole SwingUp Sparse).
>
> In principle, we can use a full trajectory, but that may not further improve performance. Recall that in infinite horizon discounted reward formulation, the discount will be so severe in further steps so that the respective stage reward plays little importance in the sum. To see that, for $\gamma = 0.99$, at N = 1000 (typical full length of an episode), $\gamma^{999} = 4e−5$. Thus, its effect is minimal. We thus consider $N = 100$ in LNSS to be effective and practical.
>
> > Q3 It would be also very interesting to study the interaction of LNSS with more gradient steps per environment step (UTD ratio) Intuitively, having more gradient updates per data collection step could also greatly help with propagating the signal back in time fast (in terms of the number of environment steps).
>
> We thank the reviewer for the suggestion. This is indeed a very interesting idea. We did not implement the UTD ratio, but implemented the original baselines approaches, i.e., one update per environment step. We did so as we thought this would be fair to all baseline algorithms by using their original implementation. But we agree with the reviewer that having more gradient updates per data collection step could help improve training time. We'd be happy to explore this idea in our future work.

---

### Official Review · Reviewer_PJwS · 2023-07-07

**Soundness:** 3 good
**Presentation:** 3 good
**Contribution:** 3 good
**Rating:** 7
**Confidence:** 3

**Summary:**

Authors proposed a long N-step surrogate stage (LNSS) reward for DRL which mitigates the high variance problem in continuous control problems. It is shown that the proposed LNSS can provide consistent performance improvement measured by average reward, convergence speed, learning success rate, and variance reduction in Q values and rewards.

**Strengths:**

Well written and well organised.
Proposed a novel and easy-to-implement n-step method LNSS which reduces variance in training. The method can be easily applied on existing algorithms such as DDPG, TD3 for enhanced performance.
Promising experimental results with large N-steps and theoretical analysis provided to support the conclusions. Nice ablation studies provided.
Main experimental results are presented in a clean table which is easy to interpret.

**Weaknesses:**

I think some interesting experimental results lacks explanation. Some figures can be improved for better readability.

**Questions:**

One question is that for baseline comparison, the authors chose original algorithms and a simple n-step variation of loss function with n=5. LNSS reported in the table is using N=100. I'm wondering can this be considered as a fair comparison? From ablation studies, smaller N such as N=5, and N=10 does not perform as good as N=100 in training and variance reduction. Can authors justify this? Would it be possible to also include LNSS N=5 in the main results table for comparison.

I noticed that in some environments, a simple n-step variation with n=5 is even much worse than the baseline algorithm, while LNSS with N=100 provide promising improvements. This is very interesting. Any explanations for this phenomenon? Will LNSS with N=5 have the same improvements?

Too many colors in figure 2 and 3, maybe use the same color for one baseline algorithm, for example, TD3-base TD3-LNSS both use red but one solid line and one dash line as you already use in the figures for an easy comparison.

A final question out of curiosity, to my knowledge, TD3 itself would fail in some complex environments , and it is nice to see how LNSS helped to improve the situation, however, some algorithms such as max-entropy algorithm SAC would succeed. Will LNSS be combined with max-entropy algorithm also provide similar improvements in performance and variance reduction?

**Limitations:**

I think for me some phenomenon in experimental results may need some further explanation. A large N is required for powerful performance.

---

> ### Author Rebuttal · Authors · 2023-08-09
>
> We greatly thank the reviewer for reading our paper and for the favorable comments. Your questions have helped us further improve the paper.
>
> >Q1 How to justify smaller N not working as well as larger N? Also how to show this in baseline comparisons (using n=5 and N=5) and ablation studies (N=5, 10, 100)?
>
> From Theorem 1 and Figure 1, we can see that for N = 5, 10, and 100, the respective $\psi$ factor is 0.2, 0.1, 0.011. Clearly, the upper bound of variance for N = 100 is much smaller than N=5 and 10. Additionally, our experiment results strongly support this theoretical result. Please refer to our discussions in Results section (Q6. Effects of different length N in LNSS) and Fig. 5. Specifically, in lines 286-289, we discussed respective learning performances for different N.
>
> As theoretical and experimental results show that the larger the N the better the learning performance. For n-step methods, as n = 5 is the best-performing choice as reported in RAINBOW, D4PG [1,2], we therefore consider it to be fair to compare with our best-performing and also realistic choice of N=100. Also since the strength of LNSS relies on using larger N, we theretofore focused on reporting large N results. The reviewer has raised an important point. If the paper is accepted, we will include N=5 in Table 1 if space permits or we will include an expanded Table 1 in the Appendix.
>
> >Q2 In some environments, a simple n-step variation with n=5 is even much worse than the baseline algorithm, while LNSS with N=100 provide promising improvements. This is very interesting. Any explanations for this phenomenon? Will LNSS with N=5 have the same improvements?
>
> Please refer to lines 256 to 259 where we explain why the performance degraded for n5 method even below the baseline. To gain some insight, from Eqn (3), about n-step methods, we can intuitively see that errors in reward signal may degrade learning performance, the longer the reward sequence, and more uncertainties due to accumulated errors from longer steps. As such, it may introduce a much larger approximation variance.
>
> For our LNSS, we do not expect LNSS with N = 5 to provide exceptionally good performance as large N values do.  Please refer to our reported results (Table 1) where N=100 improves learning performance significantly. Please also refer to our discussion for Q1 above.
>
> > Q3 Too many colors in Figures 2 and 3, maybe use the same color for one baseline algorithm, for example, TD3-base TD3-LNSS both use red but one solid line and one dash line as you already use in the figures for easy comparison.
>
> Very good point, thank you! We'll make the update if the paper is accepted.
>
> > Q4 Can LNSS work on SAC?
>
> Since our LNSS only changes the stage reward and we did not change anything else in the learning framework of baseline algorithms, theoretically LNSS can be used in SAC with no problem.
>
> Also, from DMC benchmark results [3], it is reported that SAC and TD3 have very similar performance. This is the reason we only used TD3 baseline in the paper. To provide more specifics, during the rebuttal period, we applied LNSS on SAC for two environments  (cartpole swing-up sparse and finger turn hard) to provide preliminary results. As expected, the performance of SAC+LNSS has improved over SAC baseline. Please refer to Fig R2, the preliminary results under the general response pdf file.
>
>
> [1] Hessel, M., Modayil, J., Van Hasselt, H., Schaul, T., Ostrovski, G., Dabney, W., ... & Silver, D. (2018, April). Rainbow: Combining improvements in deep reinforcement learning. In Proceedings of the AAAI conference on artificial intelligence (Vol. 32, No. 1).
>
> [2] Barth-Maron, G., Hoffman, M. W., Budden, D., Dabney, W., Horgan, D., Tb, D., ... & Lillicrap, T. (2018). Distributed distributional deterministic policy gradients. arXiv preprint arXiv:1804.08617.
>
> [3] Pardo, F. (2020). Tonic: A deep reinforcement learning library for fast prototyping and benchmarking. arXiv preprint arXiv:2011.07537.

---

> > ### Comment · Reviewer_PJwS · 2023-08-16
> > **Response to rebuttal**
> >
> > Thank the authors for their response to my questions. I'll maintain my score as accept.

---

> > > ### Author Response · Authors · 2023-08-20
> > >
> > > We sincerely thank the Reviewer for all the questions, comments, and feedback, which have helped us greatly to clarify our results and highlight our contributions. Thank you!
> > >
> > > BTW, we have included additional content in response to other Reviewers' comments. In case the Reviewer has new questions, we would be happy to address them.

---

### Official Review · Reviewer_dTq6 · 2023-07-09

**Soundness:** 1 poor
**Presentation:** 4 excellent
**Contribution:** 3 good
**Rating:** 4
**Confidence:** 5

**Summary:**

This paper proposes a simple but effective surrogate reward, which uses the future n-step cumulative rewards (with some coefficient).
The theoretical results show that the proposed method has a smaller variance than the classical one-step method.
The experimental results show the effectiveness of the proposed method.

**Strengths:**

Generally, this paper writes well.
This paper proposes an interesting surrogate reward, which is very simple but has shown to be very effective.
The paper also provides a theoretical analysis of the variance compared to that of existing one-step method.
The experimental results are complete and sufficient to show the effectiveness of the proposed method.

**Weaknesses:**

I agree that this paper tries to solve an important problem and proposes a very simple but empirically effective method.

However, the proposed method does not make sense to me. I give my reason below.

- My major concern is whether the proposed method with the surrogate reward learns exactly the same solution as the one using the original reward. The paper didn't provide any discussion or proof of this critical problem.

- In my view, my answer to this problem will be negative. Based on eq.7, the surrogate loss is, in fact, a multi-step cumulative reward times a coefficient. Compared to the existing n-step method (eq. 3), the difference is the coefficient and the next state (the proposed method use $s_t$, and the n-step method use $s_{t+N}$). The solution of the proposed method should be different from the original $Q^\pi$.

- I believe that eq. 5 is very suspicious. I don't understand why the $G'_k$ should be in that form (and $r_k'$ is a constant). In addition, in this way, the final computed $r'_k$ will be different over different k.
- (I spent a lot of hours on this paper, trying to understand, but I failed to convince myself. I'm happy to increase my score if the author can prove I'm wrong)

The paper didn't make a comparison of the variance with n-step methods.

In addition,  how does the paper fit into the off-policy setting? It is no problem for the one-step method because there is no bias. However,   when utilizing multi-step data, this could be one problem for the proposed method (DDPG and several baseline methods used in the paper are off-policy methods).

Minor Issues:
- Line 115: It's not very general to assume the policy to be deterministic.
- eq. 1: It is better to use $s_{k'+1} \sim p( | s_{k'}, a_{k'} )$. Just differentiate $k'$ and $k$
- eq. 2: missing an expectation over $s_{k+1}$.

**Questions:**

For my major concern, please see the weakness above.

---

> ### Author Rebuttal · Authors · 2023-08-09
>
> We thank the reviewer for carefully reviewing our paper and for the penetrating questions. We also thank the reviewer for recognizing that our paper addresses an important problem, and our LNSS is simple and potentially very effective.
>
> >Q1 Learns exactly the same solution as the one using the original reward?
>
> Please refer to our General Response (GR), points 5) and 6).
> The sequence ${r_k’}$ differs from ${r_k}$. Thus, the respective returns $\mathbb{Q}_i$  and $Q_i$ differ. Refer to Fig R1, the right column. Inspect estimation bias between the solid lines with the respective dashed lines. Learning with the two different reward sequences {$r_k'$} and {$r_k$} leads to different solutions in general. It is noted that LNSS resulted in significantly improved learning performance (Fig. 2 and Table 1).
>
> Note however, ${r_k’}$ approaches $r_k$ upon learning convergence, and $G_k’$ (N-step discounted sum of ${r_k’}$) becomes an unbiased estimate of $G_k$. Since the update laws are the same for both cases, and thus, at this time, the two sums necessarily converge to the same solution. But this is not true for early stage.
>
> >Q2 Eq. 5, why $G'$ is in that format and $r_k'$ is constant?
>
> Thank you for pointing this out. Please first review GR point 4). We'd like to emphasize that $r_k'$ is constructed from $G_k$ (Eqn (6)-(7)) without any assumption or additional requirement.
>
> About Eqn (5), $G_k'$ in (5) is an imaginary variable in our mind, a short-hand notation. It is probably easy to see that it inspired us to reach Eqn (6) by simply moving the sum of discounted factor $\gamma$ away from $r_k'$, and thus we arrive at Eqn (6), for the construction of $r_k'$.
>
> Our use of $G_k'$ may have caused this confusion to the reviewer. We should have used a different notation so that $G_k'$ is as in RL convention that it is the sum of discounted rewards {$r_k'$}.
>
> >Q3 comparison of the variance with n-step methods
>
> First, please note that our systematic evaluations include comparisons on variance for n-step method (n5) and LNSS, please refer to Table 1 where in all cases, the std values of LNSS are less than those of n5.
>
> Many previous works including [1-3] reported that off-policy n-step methods such as TD$(\lambda)$ and $Q(\sigma)$ are shown to decrease the estimation bias at the cost of increasing the variance.
> As LNSS aims to, and is shown to, reduce variance without sacrificing bias (General Response Point 7), to see how n-step methods may increase variance, please refer to Eqn (25), the n-step return variance can be written as $ Var[\sum_{t=k}^{k+N-1}\gamma^{t-k} r_t ] \leq (\frac{\gamma^{2N} - 1}{\gamma^2 - 1}) \mathbb{B} $ and $\frac{\gamma^{2N} - 1}{\gamma^2 - 1}) > 1$ which will result in the upper bound on the variance of n-step to be greater than that of single step. Combine with Theorem 1 we will have the following relations hold $Var[$LNSS$] < Var[$single step$] < Var[$n step$]$.
>
> >Q4 About estimation bias and off-policy issue
>
> Please refer to our General Response, points 7)-9).
>
> > Minor Issues
>
> We thank the reviewer for pointing out these detailed discrepancies. We will make the edits if the paper is accepted.
>
> [1] Watkins,C.J.C.H.(1989).Learning from delayed rewards.PhD Thesis,University of Cambridge, England.
>
> [2] Jaakkola, T.; Jordan, M. I.; and Singh, S. P. 1994. On the convergence of stochastic iterative dynamic programming algorithms. Neural Computation 6(6):1185–1201.
>
> [3] De Asis, Kristopher, J. Hernandez-Garcia, G. Holland, and Richard Sutton. "Multi-step reinforcement learning: A unifying algorithm." In Proceedings of the AAAI Conference on Artificial Intelligence, vol. 32, no. 1. 2018.

---

> > ### Comment · Reviewer_dTq6 · 2023-08-11
> > **Furthre question**
> >
> > I thank the authors for their feedback.
> > Could you please provide strict theorems and proof that showing the proposed method is unbiased with $Q^*$ or $Q^\pi$?
> >
> > I checked your response about the unbiased property in points 7)-9). I failed to see these results theoretically. Most of them are discussions and it's hard to identify which parts are assumptions and which parts are conclusions. It would be more clear if the authors could give a strict theorem and clearly state all the conditions, and the results we can have, it will be more convincing.
> >
> > My answer to this problem is still negative yet. I don't believe that this method can be unbiased (the reasons have been given in my first review).
> > I believe that this is very important because we need to carefully understand what we are doing and what is meaning of the results.

---

> > > ### Author Response · Authors · 2023-08-20
> > > **Central Questions Reiterated For Clarification & Calibration**
> > >
> > > We greatly appreciate the Reviewer’s comments and feedback. We thank the Reviewer for recognizing that the paper “tries to solve an important problem and proposes a very simple but empirically effective method.” We also agree that the questions/comments from the Reviewer are “very important because we need to carefully understand what we are doing and what is meaning of the results.”
> > >
> > > First, we’d like to make sure that we’re on the same page with the Reviewer by re-iterating the issues that the Reviewer has raised. The central questions are:
> > >
> > > >**Central Question1)**	(from the original review comments) Does the proposed method with the surrogate reward learns exactly the same solution as the one using the original reward. The Reviewer considers that “the solution of the proposed method should be different from the original $Q^\pi$.”
> > >
> > > >**Central Question 2)**	(from the comments to our Rebuttal) (Theoretically) “Please provide strict theorems and proof that showing the proposed method is unbiased with $Q^*$ or $Q^\pi$“ as the Reviewer checked points 7)-9) in Rebuttal but failed to see these results. The Reviewer does not “believe that this method can be unbiased (the reasons have been given in my first review).”
> > >
> > > Addressing these questions requires clarifications of some key terms that the Reviewer used in the questions. Below, we elaborate on possible interpretations of these terms, and then we provide respective answers.
> > >
> > > Before we proceed, we would like to define the following for the ease of discussion.
> > >
> > > $\bullet$ $Q$ is for action value associated with the original reward sequence $r_k$.
> > >
> > > $\bullet$ $\mathbb{Q}$ is for action value associated with the surrogate reward sequence $r'_k$.
> > >
> > > $\bullet$ $\pi_j$ is the policy at the $j$th iteration. For learning using $r'_k$, we use $\bar{\pi}_j$ instead.
> > >
> > > In the upcoming windows, we will address these central questions according to two categories of possible interpretations of some key terms:
> > >
> > > **1) What does the Reviewer mean by "$Q^\pi$". (Answers are provided in Case 1.1 and Case 1.2)**
> > >
> > > **2) What does the Reviewer mean by "Bias".(Answers are provided in Case 2.1, Case 2.2, and Case 2.3)**

---

> > > > ### Author Response · Authors · 2023-08-20
> > > > **Case 1.1 & Case 1.2, to Address What the Reviewer May Mean by $Q^\pi$**
> > > >
> > > > >	What does the Reviewer mean by ”$Q^\pi$”?
> > > >
> > > >
> > > > > **Case 1.1)** $\pi$ refers to a given policy, $Q^\pi$ is an evaluation of this policy for the original reward $r_k$, and $\mathbb{Q}^\pi$ for surrogate reward $r'_k$ from LNSS. The Reviewers wants to know if the two evaluations are the same, i.e., if  $Q^\pi = \mathbb{Q}^\pi$.
> > > >
> > > > We give the following theorem.
> > > >
> > > > **Theorem R1.1** Assume that the reward $r_k$ is bounded within $[0, R_{max}] (Rmax >0)$. We also assume that the given policy $\pi$ results in the same expectation of stage reward , i.e.,
> > > > $\mathbb{E}^\pi[r_k] = \mathbb{E}^\pi[r'_k]$ for all $k$. Then $Q^\pi$ using original reward and $\mathbb{Q}^\pi$ using LNSS are the same, i.e.,
> > > >
> > > > $\mathbb{Q}^\pi(s_k,a_k) = Q^\pi(s_k,a_k).  $    --- Eqn (R1)
> > > >
> > > >
> > > > Proof.
> > > >
> > > > Given $\mathbb{E}^\pi[r'_k]  =  \mathbb{E}^\pi[r_k]$ for all $k$. Since
> > > >
> > > > $Q^\pi(s_k,a_k) = \mathbb{E}^\pi[\sum_{t=k}^{\infty} \gamma^{t-k} r_t] $  --- Eqn (R2)
> > > >
> > > > $\mathbb{Q}^\pi(s_k,a_k) = \mathbb{E}^\pi[\sum_{t=k}^{\infty} \gamma^{t-k} r'_t] $  --- Eqn (R3)
> > > >
> > > > By applying $\mathbb{E}^\pi[r'_k] = \mathbb{E}^\pi[r_k]$ to Eqn (R2) and Eqn (R3).
> > > >
> > > > $\mathbb{Q}^\pi(s_k,a_k) = Q^\pi(s_k,a_k)$  --- Eqn (R4)
> > > >
> > > > Thus Theorem R1.1 holds.
> > > >
> > > > **Remark R1.1.**
> > > >
> > > > $\bullet$ To shed some light on the assumption of expectation of stage reward, consider the constant stage reward example we provided in the Rebuttal. In this case,
> > > > $\mathbb{E}^\pi[r_k] = C$ for all $k$. By apply Eqn (7) in the paper,
> > > >
> > > > $\mathbb{E}^\pi[r'_k]  = $
> > > >
> > > > $ \mathbb{E}^\pi[\frac{\gamma - 1}{\gamma^N - 1}\sum_{t=k}^{k+N-1} \gamma^{t-k} r_t] =C$.
> > > >
> > > > From Theorem R1.1, $ \mathbb{Q}^\pi(s_k,a_k) = Q^\pi(s_k,a_k)$.
> > > >
> > > > $\bullet$ However if $\mathbb{E}^\pi[r_k] \neq \mathbb{E}^\pi[r'_k]$ for any $k$, then from Eqn (R2) and Eqn (R3), $\mathbb{Q}^\pi(s_k,a_k) \neq Q^\pi(s_k,a_k)$.
> > > > >**Case 1.2)**  $\pi$ refers to policy $\pi_j$ for $j \to {\infty}$. Then in this case, the Reviewer considers
> > > > $Q^\pi=Q^{\pi_{\infty}}$, $\mathbb{Q}^{\bar{\pi}}=\mathbb{Q}^{\bar{\pi}_{\infty}}$. The Reviewer wants to know if  $Q^{\pi}$= $\mathbb{Q}^{\bar{\pi}}$.
> > > >
> > > >
> > > > In this case, we have $Q^{\pi_{\infty}} \neq \mathbb{Q}^{\bar{\pi}_{\infty}}$. This is because using different rewards, the target policy will be different during policy improvement so that it results in different $Q$ values.
> > > >
> > > >
> > > > **Remark R1.2.**
> > > >
> > > > This difference ($Q^{\pi_{\infty}} - \mathbb{Q}^{\bar{\pi}_{\infty}}$) between different algorithms or methods commonly exists in RL. For example, SAC [R1] augments the objective with an expected entropy of policy
> > > >
> > > >  $\pi$ as $J(\pi)=\mathbb{E}^\pi [\sum_{t=k}^\infty \gamma^{t-k}(r_k+\alpha \mathcal{H}(\pi(\cdot \mid \mathbf{s}_k)))]$.
> > > >
> > > >   It is possible that $\mathbb{E}^{\pi}[r_k] \neq \mathbb{E}^{\pi}[r_k+\alpha \mathcal{H}(\pi(\cdot \mid \mathbf{s}_k))]$
> > > >
> > > > which leads to $Q_{SAC}(s_k,a_k) \neq Q(s_k,a_k)$. Consider another case of D4PG [R2]. It aims at learning a distribution whose expectation equals to $Q$, namely $Q_{D4PG}(s_k,a_k) = E[Z_\pi(s_k,a_k)]$. In D4PG, there is a required bound on the value function. This hyperparameter directly affects the value of the $Q_{D4PG}(s_k,a_k)$ so that it is possible that $Q_{D4PG}(s_k,a_k) \neq Q(s_k,a_k)$.
> > > >
> > > > Furthermore, the reward estimation methods mentioned in our paper [R3-R10] will all have this difference since they use models (either in the form of neural networks, environment transitions, or reward noise models) to estimate the true stage reward which can result in model estimation error so that the respective $Q$ values will be different from using original reward.

---

> > > > > ### Author Response · Authors · 2023-08-20
> > > > > **Case 2.1 & Case 2.2, to Address What the Reviewer May Mean by Bias**
> > > > >
> > > > > >What does the Reviewer mean by “bias”?
> > > > >
> > > > > >**Case 2.1)** The bias considered by the Reviewer is the difference $\delta$ = $Q^*- \mathbb{Q}^*$.
> > > > >
> > > > > In this case, the analysis follows from that for Case 1.1) along with the same assumption on reward $r_k$. Assume that the 1) reward $r_k$ is bounded by $[0, R_{max}], R_{max} \geq 0$ and 2) the two optimal policies are the same $\pi^*$ that results in  $\mathbb{E}^{\pi^*}[r_k] = \mathbb{E}^{\pi^*}[r'_k]$ for all $k$. Then from Theorem R1.1, $\mathbb{Q}^*(s_k,a_k) = Q^*(s_k,a_k)$. However, if $\mathbb{E}^{\pi^*}[r_k] \neq \mathbb{E}^{\pi^*}[r'_k]$ for any $k$, then from Eqn (R2) and Eqn (R3), $\mathbb{Q}^*(s_k,a_k) \neq Q^*(s_k,a_k)$.
> > > > >
> > > > > > **Case 2.2)** The reviewer considers that the bias = $\mathbb{Q}^* - \mathbb{Q}^{\bar{\pi}^{*}}$,
> > > > >
> > > > > > where theoretically $\bar{\pi}^* = \bar{\pi}_{\infty}$.
> > > > >
> > > > >
> > > > > Theoretically, LNSS will have the same property as those used for the original reward. By using $r'_k$ we have an unbiased estimate with its own optimal value upon learning convergence. Proof now follows.
> > > > >
> > > > >
> > > > > **Lemma R2.1** (LNSS Policy Evaluation). Consider stage reward $r_k$ being bounded. Then the sequence $\mathbb{Q}_i$ converges to $\mathbb{Q}^{\bar{\pi}}$ as $i \to \infty$.
> > > > >
> > > > > Proof. Define the LNSS reward as $r_k' = \frac{\gamma - 1}{\gamma^N - 1}\sum_{t=k}^{k+N-1} \gamma^{t-k} r_t$ and
> > > > > consider the LNSS Bellman backup operator as
> > > > >
> > > > > $\mathbb{Q}_{i+1}(s_k,a_k) =r_k' + \gamma \mathbb{E}^{\bar{\pi}}[\mathbb{Q}_i (s_k+1, a_k+1)].$  --- Eqn (R5)
> > > > >
> > > > > Since LNSS is only substituting the original reward $r_k$ by LNSS reward $r_k'$, applying the standard convergence results for policy evaluation [R1, R12] leads to Lemma R2.1. (Specifically Section 4.1).
> > > > >
> > > > > **Please note**, this makes LNSS advantageous over other estimation methods ([R3-R10]), which require models to perform reward estimation or reward surrogation which in real implementations will contribute to estimation bias. Instead, LNSS collects samples along the current behavior policy which does not require any model and thus no model estimation error.
> > > > >
> > > > > **Lemma R2.2** (LNSS Policy Improvement). Let $\bar{\pi}_j \in \Pi$ be the policy at iteration $j$,
> > > > >
> > > > > $\bar{\pi}_{j+1}(s_k)$
> > > > >
> > > > > $= \arg \max_{a_k} \mathbb{Q}^{\bar{\pi}_j}(s_k,a_k)$.
> > > > >
> > > > > Then $\mathbb{Q}^{\bar{\pi}_{j+1}}(s_k,a_k)$
> > > > >
> > > > > $ \geq \mathbb{Q}^{\bar{\pi}_{j}}(s_k,a_k)$ for all $(s_k,a_k) \in \mathbf{S} \times \mathbf{A}$.
> > > > >
> > > > > Proof. It can be shown (see [R12]) by applying the standard policy improvement theorem. (Specifically Section 4.2).
> > > > >
> > > > > **Theorem R2.2** Repeated application of policy evaluation and policy improvement to policy $\bar{\pi} \in \Pi$ results in convergence to a policy $\bar{\pi}^*$ such that $ \mathbb{Q}^{\bar{\pi}^*}(s_k,a_k) \geq \mathbb{Q}^{\bar{\pi}}(s_k,a_k)$ for all $\bar{\pi} \in \Pi$ and all $(s_k,a_k) \in \mathbf{S} \times \mathbf{A}$. This implies that $ \mathbb{Q}^*(s_{k},a_{k}) - \mathbb{Q}^{\bar{\pi}^*}(s_{k},a_{k}) = 0$.
> > > > >
> > > > > Proof.  Let $\bar{\pi}_j$ be the policy at iteration $j$.
> > > > >
> > > > > By Lemma R2.2, the sequence $\mathbb{Q}^{\bar{\pi}_j}(s_k,a_k)$ is monotonically increasing.
> > > > >
> > > > > Since $\mathbb{Q}$ is bounded from above for $\bar{\pi} \in \Pi$, the $\mathbb{Q}^{\bar{\pi}_j}(s_k,a_k)$ will converges to some $\mathbb{Q}^{\bar{\pi}^*}(s_k,a_k)$.
> > > > >
> > > > >  Next we show that $\bar{\pi}^*$ is optimal. At convergence, it must be the case that $\mathbb{Q}^{\bar{\pi}^*}(s_k,\bar{\pi}^*(s_k)) \geq \mathbb{Q}^{\bar{\pi}}(s_k,\bar{\pi}(s_k))$ for any $\bar{\pi} \in \Pi$ and $\bar{\pi} \neq \bar{\pi}^*$. On the other hand, for all $\mathbb{Q}$ values corresponding to any other non-optimal policy, the optimal $Q$ value has the following, $\mathbb{Q}^*(s_k,a_k) \geq \mathbb{Q}(s_k,a_k)$ for all $(s_k,a_k) \in \mathbf{S} \times \mathbf{A}$. Hence $\bar{\pi}^*$ is optimal in $\Pi$ and we have $ \mathbb{Q}^*(s_{k},a_{k}) = \mathbb{Q}^{\bar{\pi}^*}(s_{k},a_{k})$ , i.e.,  $ \mathbb{Q}^*(s_{k},a_{k}) - \mathbb{Q}^{\bar{\pi}^*}(s_{k},a_{k}) = 0$.

---

> > > > > > ### Author Response · Authors · 2023-08-20
> > > > > > **Case 2.3, to Address What the Reviewer May Mean by Bias**
> > > > > >
> > > > > > >**Case 2.3)** The Reviewer considers that the bias is what is in the main stream SOTA literature, that is, the bias is as defined in Eqn (R6) below.
> > > > > >
> > > > > > In this case, we agree with the Reviewer that it is important to understand algorithmic properties, especially in the current context. As in the literature [R1, R11, R12, R13, R16], we characterize this "bias" by the following measure,
> > > > > >
> > > > > > $\delta = \mathbb{E}^\pi[\sum_{t=k}^{\infty} \gamma^{t-k}r_t|s_k,a_k] -  Q_{i}(s_{k},a_{k}).$   --- Eqn (R6)
> > > > > >
> > > > > > Unlike Case 2.1 and Case 2.2, where conclusions can be drawn under ideal conditions, the current case is about what happens in reality if we consider various errors that contribute to the bias measure (Eqn. R6). Consequently,  all methods we discussed in our paper are biased, including LNSS based methods as the Reviewer interpreted.
> > > > > >
> > > > > > To shed more light, please consider the following perspectives which help characterize the "bias" issue.
> > > > > >
> > > > > > 1) Let the optimal value function $Q^*$ correspond to the unique solution of the Bellman optimality equation. The well-established $Q$-learning algorithms perform value update based on the Bellman optimality operator.  Such iterative value updates are proved to converge to the optimal value function $Q^*$ under some properly stated conditions. However, it is well known that
> > > > > > "overestimation bias is a property of $Q$-learning" [R13] in reality this is caused by the maximization of a noisy value estimate which induces a consistent overestimation [R14]. As another example, TD3 and SAC both use double critic networks to address the overestimation bias. In turn, it may induce underestimation bias [R1,R13].
> > > > > >
> > > > > > 2) A major source of estimation bias (Eqn R6) comes from the optimization procedure. Even though we have theoretical guarantees of sufficient expressiveness of representation by neural networks to accurately represent value function, exactly solving this optimization problem is hard. The mini-batch gradient update may cause unpredictable effects on state-action pairs that are inevitably outside the training batch [R12, R17, R18]. The variances in gradient estimates in the ubiquitous stochastic gradient descent would inevitably result in approximation errors in value estimation.
> > > > > >
> > > > > > 3) As said in [R11], "Overestimation bias, however, may not always be detrimental since it encourages exploration for overestimated actions. In some cases, however, erring towards an underestimation bias can be harmful since underestimation bias might discourage exploration." But if the estimated value is much off the true value from either direction, it definitely hurts the performance.
> > > > > >
> > > > > > 4)  The estimation bias alone is not a sufficient measure of algorithm performance. As in D4PG, DDPG, TD3, and SAC [R1,R2,R13,R15], the average reward $\frac{1}{E} \sum_{n=0}^{E}\sum_{t=0}^{T}r_{n_t}$ and its variance are often used in different environments to evaluate algorithm performances.
> > > > > >
> > > > > > 5) Along the same line, for LNSS, we have evaluated the estimation error according to Eqn (R6) (Fig. R1). We can see a clear improvement in average reward and variance reduction. In the meantime, upon convergence, LNSS does not introduce extra errors from its base algorithms.
> > > > > >
> > > > > > 6) In summary, $Q^*$ is rarely achieved in reality in RL. If two methods have exactly the same solution of $Q$, then there is no performance improvement per se. For example, comparing with the original Bellman equation (as in DDPG), TD3 changes target to
> > > > > > $y = r_k + \gamma \min_{i = 1,2} Q(s_{k+1},a_{k+1})$, and SAC uses yet another target  $y = r_k + \gamma V(s_{k+1})$, where $V(s_{k}) = E_{a_k \sim p_\pi} [Q(s_{k},a_{k}) - log \pi (a_k|s_k)]$. They converge to different solutions and thus, we can tell some algorithms are better than others.
> > > > > >
> > > > > > 7) Finally we conclude that LNSS-based methods, as other popular algorithms, are biased in the context of Case 2.3. Our extensive results in our paper (Figure 2-6 and Table 1) show that LNSS has significantly improved performance in terms of average reward, learning speed, success rate, and reduced variance.

---

> > > > > > > ### Author Response · Authors · 2023-08-20
> > > > > > > **References**
> > > > > > >
> > > > > > > [R1] Haarnoja, T., Zhou, A., Abbeel, P., \& Levine, S. (2018, July). Soft actor-critic: Off-policy maximum entropy deep reinforcement learning with a stochastic actor. In International conference on machine learning (pp. 1861-1870). PMLR.
> > > > > > >
> > > > > > > [R2] Barth-Maron, G., Hoffman, M. W., Budden, D., Dabney, W., Horgan, D., Tb, D.,  \& Lillicrap, T. (2018). Distributed distributional deterministic policy gradients. arXiv preprint arXiv:1804.08617.
> > > > > > >
> > > > > > > [R3] Gabriel Dulac-Arnold, Nir Levine, Daniel J Mankowitz, Jerry Li, Cosmin Paduraru, Sven Gowal, and Todd Hester. Challenges of real-world reinforcement learning: definitions, benchmarks and analysis. Machine Learning, 110(9):2419–2468, 2021.346
> > > > > > >
> > > > > > > [R4] Johan Bjorck, Carla P Gomes, and Kilian Q Weinberger. Is High Variance Unavoidable in RL? A Case Study in Continuous Control. arXiv preprint arXiv:2110.11222, 2021.
> > > > > > >
> > > > > > > [R5] Joshua Romoff, Peter Henderson, Alexandre Piché, Vincent Francois-Lavet, and Joelle Pineau. Reward estimation for variance reduction in deep reinforcement learning. arXiv preprint arXiv:1805.03359, 2018.
> > > > > > >
> > > > > > > [R6] Jingkang Wang, Yang Liu, and Bo Li. Reinforcement learning with perturbed rewards. In Proceedings of the AAAI conference on artificial intelligence, volume 34, pages 6202–6209, 2020.
> > > > > > >
> > > > > > > [R7] Harm Van Seijen and Richard S Sutton. Efficient planning in MDPs by small backups. In Proc.30th Int. Conf. Mach. Learn, volume 28. Citeseer, 2013.407
> > > > > > >
> > > > > > > [R8] Stefan Depeweg, José Miguel Hernández-Lobato, Finale Doshi-Velez, and Steffen Udluft. Learning and policy search in stochastic dynamical systems with bayesian neural networks. arXiv preprint arXiv:1605.07127, 2016.
> > > > > > >
> > > > > > > [R9] David Silver, Hado Hasselt, Matteo Hessel, Tom Schaul, Arthur Guez, Tim Harley, Gabriel Dulac-Arnold, David Reichert, Neil Rabinowitz, Andre Barreto, et al. The predictron: End-to-end learning and planning. In International Conference on Machine Learning, pages 3191–3199. PMLR, 2017.
> > > > > > >
> > > > > > > [R10] Vladimir Feinberg, Alvin Wan, Ion Stoica, Michael I Jordan, Joseph E Gonzalez, and Sergey Levine. Model-based value estimation for efficient model-free reinforcement learning. arXiv preprint arXiv:1803.00101, 2018.
> > > > > > >
> > > > > > > [R11] Lan, Q., Pan, Y., Fyshe, A., \& White, M. (2020). Maxmin q-learning: Controlling the estimation bias of q-learning. arXiv preprint arXiv:2002.06487.
> > > > > > >
> > > > > > > [R12] Sutton, R. S., \& Barto, A. G. (2020). Reinforcement learning: An introduction. MIT press.
> > > > > > >
> > > > > > > [R13] Fujimoto, S., Hoof, H., \& Meger, D. (2018, July). Addressing function approximation error in actor-critic methods. In International conference on machine learning (pp. 1587-1596). PMLR.
> > > > > > >
> > > > > > > [R14] Thrun, S., \& Schwartz, A. (2014, March). Issues in using function approximation for reinforcement learning. In Proceedings of the 1993 connectionist models summer school (pp. 255-263). Psychology Press.
> > > > > > >
> > > > > > > [R15] Lillicrap, T. P., Hunt, J. J., Pritzel, A., Heess, N., Erez, T., Tassa, Y., ... \& Wierstra, D. (2015). Continuous control with deep reinforcement learning. arXiv preprint arXiv:1509.02971.
> > > > > > >
> > > > > > > [R16] Chen, Xinyue, et al. "Randomized ensembled double q-learning: Learning fast without a model." arXiv preprint arXiv:2101.05982 (2021).
> > > > > > >
> > > > > > > [R17] Riedmiller, Martin. "Neural fitted Q iteration–first experiences with a data efficient neural reinforcement learning method." Machine Learning: ECML 2005: 16th European Conference on Machine Learning, Porto, Portugal, October 3-7, 2005. Proceedings 16. Springer Berlin Heidelberg, 2005.
> > > > > > >
> > > > > > > [R18]  Hado Van Hasselt, Yotam Doron, Florian Strub, Matteo Hessel, Nicolas Sonnerat, and Joseph Modayil. Deep reinforcement learning and the deadly triad. arXiv preprint arXiv:1812.02648,2018.

---

### Author Rebuttal · Authors · 2023-08-07

# General Response: Insights on LNSS, its essence, and novelty
We first thank all reviewers full-heartedly for their insightful comments. We will integrate some key points below into the revision if our paper is accepted.
1) LNSS is about a new, surrogate reward $r_k’$, which relies on multiple (long N) steps of the reward sequence {$r_k$}. This new surrogate reward sequence {$r_k’$} is then used in a 1-step framework (Eqn 2).
2) LNSS is new as it is fundamentally different from 1) the traditional 1-step methods (Eqn 2) due to difference in stage reward, and 2) the traditional multi-step or n-step methods (Eqn 3) due to using a single stage surrogate reward $r_k’$ (Eqn 8). Both traditional approaches have a long history of development [1-4]
3) LNSS is simple, and effective as it results in reduced variance and improved performance (average reward, convergence speed, learning success rate).
4) The surrogate $r_k’$ is a weighted average (or simply the center of mass) of the long N-step reward sequence as
$ r_k' =  \dfrac{ \sum_{t=k}^{k+N-1} \gamma^{t-k} r_t}{\sum_{n=0}^{N-1} \gamma^{n}} = \dfrac{G_k}{ \dfrac{\gamma^N - 1}{\gamma -1}}=  \dfrac{\gamma -1}{\gamma^N - 1}G_k$
5) The sequence {$r_k’$} differs from {$r_k$}. Thus, their respective $\mathbb{Q}_i$ and $Q_i$ differ. Refer to Figure R1, the right column. Inspect estimation bias between the solid lines with the respective dashed lines. Notice that, the two estimation biases differ during learning. However, as will be discussed below, $r_k’$ approaches $r_k$ upon learning convergence, at which point $G_k’$ as the N-step discounted sum of $r_k’$ (not the $G_k'$ in Eqn (5), where we should have used a different shorthand notation) become an unbiased estimate of $G_k$.
6) To see the above point, notice that, for successful learning using either {$r_k’$} or {$r_k$} (namely reaching convergence with sufficiently large $i$ and sample pairs being used sufficient number of times in learning), it is necessary that $r_k$ approaches a constant reward $c$ (see DMC examples after the below equations). From the derivations below, if we compute $r_k’$ as constructed above or Eqn (6), it is necessary that $r_k’$ also approaches $c$. Since $r_k’$ approaches $r_k$ (constant $c$) upon learning convergence, and the update laws are the same for both cases, then, toward late stage of learning, $G_k$ and $G_k'$ necessarily converge to the same solution, but this is not true for early stage.

    Derivation: For successful learning, $r_t = c$, $G_k = \sum_{t=k}^{k+N-1} \gamma^{t-k} r_t =  \dfrac{1 - \gamma^N}{1 - \gamma}c$, thus For Eqn (6), $r_k' = \dfrac{\gamma -1}{\gamma^N - 1}G_k = c$ and $G_k' = \sum_{t=k}^{k+N-1} \gamma^{t-k} r'_t =  \dfrac{1 - \gamma^N}{1 - \gamma}c$

    For illustration, consider a DMC example: a two-link planar reacher where the stage reward approaches 1 as end point approaches target. Thus, the reward sequence {$r_k$} will be all 1’s once learning converges. As another example from DMC, consider the “acrobot swingup” which involves an underactuated double pendulum environment. Once the double pendulum is balanced, and then to the end of this episode, the reward sequence {$r_k$} will be all 1’s.

7) From the above discussion, upon learning convergence, $G_k’$ is an unbiased estimate of $G_k$. Also, from Fig R1 in attached PDF, we see that LNSS does not introduce additional estimation bias to the base algorithms (inspect the right column, late stage of learning).
8) We speculate that the initial difference between $G_k’$ and $G_k$ may actually be helpful to learning performance with reduced variance, improved average reward, learning success rate, and learning speed (Refer to Fig 2, and Table 1). Or in other words, using knowledge of the ultimate expected return ($G_k$) early on may have helped steer certain transitions and hence shape the behavior of the agent.
9)	$r_k’$ is constructed from a segment of N-steps of the reward sequence given policy (Eqn 7). It is then placed into off-policy baseline algorithms (e.g., DDPG,D4PG, TD3, SAC) in a single step framework (Eqn 2). Also notice that, as discussed in the above point 7), using $r_k’$ does not introduce additional estimation bias. Thus together (LNSS on top of baseline), the procedure still is an off-policy TD algorithm and the learned action-value function still aims at approximating $Q^*$, independent of the policy being followed, so long as the policy has an effect in that it determines which state–action pairs are visited and updated, and all state-action pairs continue to be updated. This a minimal requirement for any method guaranteed to find optimal behavior in the general case [Sutton 2018 book].
10)	LNSS theoretically reduces the upper bound of variance according to Theorem 1. Furthermore, this bound exponentially decreases as N increases. For example, for N=100, Refer to Remark 2 and Fig 1, we can see that $\psi=0.011$. Therefore, in this case, LNSS has reduced the upper bound of variance by about 100 times from single-step methods. Our extensive evaluation results also show significantly reduced variance due to LNSS. (Table 1 and Fig 4, Fig 6 and all figures in Appendix)

[1] Watkins,C.J.C.H.(1989).Learning from delayed rewards.PhD Thesis,University of Cambridge, England.

[2] Jaakkola, T.; Jordan, M. I.; and Singh, S. P. 1994. On the convergence of stochastic iterative dynamic programming algorithms. Neural Computation 6(6):1185–1201.

[3] De Asis, Kristopher, J. Hernandez-Garcia, G. Holland, and Richard Sutton. "Multi-step reinforcement learning: A unifying algorithm." In Proceedings of the AAAI Conference on Artificial Intelligence, vol. 32, no. 1. 2018.

[4] Mnih, V., Badia, A. P., Mirza, M., Graves, A., Lillicrap, T., Harley, T., ... \& Kavukcuoglu, K. (2016, June). Asynchronous methods for deep reinforcement learning. In International conference on machine learning (pp. 1928-1937). PMLR.

---

### Decision · Program_Chairs · 2023-09-21

**Decision:**

Accept (poster)

**Comment:**

This paper has initially received mixed reviews. Some reviewers have appreciated the paper, while others have expressed doubts about the soundness of the method. In particular, one reviewer seems really concerned about whether the proposed method introduces bias or not, as claimed. Unfortunately, even if the authors provided an extensive rebuttal to address the concerns of reviewers in terms of the contribution and soundness of the method, the reviewers with negative scores did not participate in the discussion till the end. Given the positive reviews and the effort of the authors in providing theoretical guarantees for their method, even for multiple cases trying to interpret the doubts of the reviewer, I argue for accepting this paper.

I encourage the authors to address all the comments and to incorporate the recommended improvements in the final version.